# PAIRBENCH: ARE VISION-LANGUAGE MODELS RELIABLE AT COMPARING WHAT THEY SEE?

## ABSTRACT

Understanding how effectively large vision language models (VLMs) compare visual inputs is crucial across numerous applications, yet this fundamental capability remains insufficiently assessed. While VLMs are increasingly deployed for tasks requiring comparative judgment, including automated evaluation, re-ranking, and retrieval-augmented generation, no systematic framework exists to measure their performance in these scenarios. We present **PAIRBENCH**, a simple framework that evaluates VLMs as customizable similarity tools using widely available image datasets. Our approach introduces four key metrics for reliable comparison: alignment with scores derived from human annotations, consistency across pair ordering, distribution smoothness, and controllability through prompting. Our analysis reveals that no model consistently excels across all metrics, with each demonstrating distinct strengths and weaknesses. Most concerning is the widespread inability of VLMs to maintain symmetric similarity scores. Interestingly, we demonstrate that performance on our benchmark strongly correlates with popular benchmarks used for complex reasoning tasks, while providing additional insight into controllability, smoothness and ordering. This makes **PAIRBENCH** a unique and comprehensive framework to evaluate the performance of VLMs for automatic evaluation, while offering an efficient predictor of model capabilities for more complex tasks. Our evaluation code and dataset are available to the research community at `https://anonymous.4open.science/r/pairbench-6C08`.

## 1 INTRODUCTION

Vision language models (VLMs) have progressed to the point of having impressive performance on a wide array of tasks (Achiam et al., 2023; Laurençon et al., 2024; Reid et al., 2024; Abdin et al., 2024; Wang et al., 2024c; Grattafiori et al., 2024), ranging from summarization and visual question answering to image captioning and common sense reasoning (Kembhavi et al., 2016; Johnson et al., 2017; Zellers et al., 2019; Lu et al., 2023; Chen et al., 2024b; Liu et al., 2025; Kazemi et al., 2024; Kil et al., 2024). While human evaluation remains the gold standard for assessing model outputs, it is expensive, time-consuming, and prone to inconsistency due to annotator variance (Liu et al., 2019; Knox et al., 2024; Feng et al., 2024). Consequently, practitioners increasingly deploy more powerful VLMs as automated evaluators across diverse applications including model assessment, content ranking, and information retrieval systems (Mañas et al., 2024; Liu et al., 2024a; 2025).

The efficacy of VLMs in these comparative tasks fundamentally depends on their ability to function as reliable similarity kernels, consistently measuring the relevance between data pairs regardless of context. However, this critical capability remains insufficiently examined. Current evaluation approaches either fail to isolate comparison abilities or require expensive expert validation, and little to no guidance exists when selecting models for comparison-dependent tasks. As illustrated in Figure 1, even widely used and highly capable commercial models like `GPT-4o-1120` and `Gemini-1.5-Pro` demonstrate concerning inconsistencies when comparing visual inputs, sometimes failing to follow similarity assessment instructions or producing asymmetric scores for identical pairs presented in different orders, which exemplifies the extent to which evaluation of comparison skills are lacking.

To address this gap, we introduce **PAIRBENCH**, a framework designed to evaluate VLMs as similarity estimators using readily available datasets and straightforward transformation techniques. Our approach optimizes the signal-to-evaluation cost ratio by focusing on four essential metrics: **MMScore**

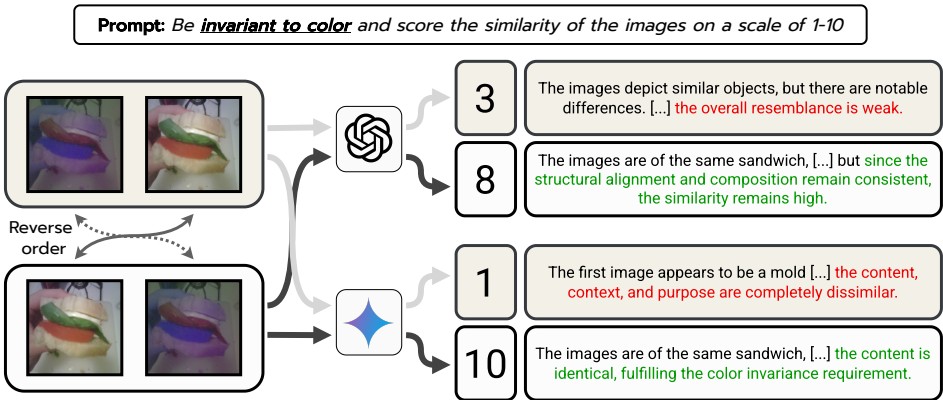

Figure 1: Image order change; prompting `GPT-4o-1120` and `Gemini-1.5-Pro` with identical text and image prompts, differing only in image order, leads to varying predicted scores. *Auto evaluators defined by these models will yield drastically different judgments after minor changes in the prompt.* Detailed failure cases of state-of-the-art models are reported in Appendix A.

(alignment with human judgment), $\varepsilon$-**RelaxSym** (consistency across pair ordering), **Smoothness** (distribution of scores), and **Controllability** (response to prompt instructions). We instantiated PAIRBENCH using easily accessible datasets. Namely, ImageNet (Deng et al., 2009), MS-COCO (Lin et al., 2014), and WhatsUp (Kamath et al., 2023) seeded the evaluation suite we built. We further conducted a human study to establish ground truth similarity scores that enable direct measurement of how well model assessments align with human perception. By applying controlled transformations to create synthetic paired images with specific feature differences, PAIRBENCH enables precise examination of model biases and strengths in detecting various types of visual differences.

We carried out an extensive evaluation covering several state-of-the-art VLMs, both proprietary and open-source, multiple dataset configurations, and different prompt templates. Results reveal not only significant variations in comparison capabilities across different architectures and training approaches, but also show concerning asymmetries in how models process the same data pairs when presented in different orders, and highlight which models can be effectively controlled through prompt instructions. Remarkably, despite its simplicity, the performance on PAIRBENCH strongly correlates with results on complex reasoning benchmarks (Yue et al., 2024; Lu et al., 2023; Chen et al., 2024b; Guan et al., 2024; Liu et al., 2024b; Kembhavi et al., 2016), suggesting that many advanced tasks ultimately rely on models functioning as effective similarity kernels.

Our contributions are as follows:

- We propose **PAIRBENCH**, a framework for evaluating VLMs as similarity kernels, which does not require additional expert annotations and is cheap to instantiate.

- We further create and release[1] four instantiations of PAIRBENCH using popular datasets - ImageNet, MS-COCO, and WhatsUp - which consist of 70K data pairs for comparisons.

- We carry out a broad benchmarking of several closed- and open-source VLMs on the different configurations within our proposed dataset instantiations to show how models differ and give insight into what extent they can be trusted to act as auto evaluators on image-image and image-text data pairs.

- Lastly, we report the correlations of the results of our framework with popular benchmarks and show the ability to compare, captured by the metrics in PAIRBENCH, have predictive power of performance on several tasks and can act as a low-cost surrogate during training or validation of VLMs.

---

[1]https://huggingface.co/datasets/feiziaarash/pairbench

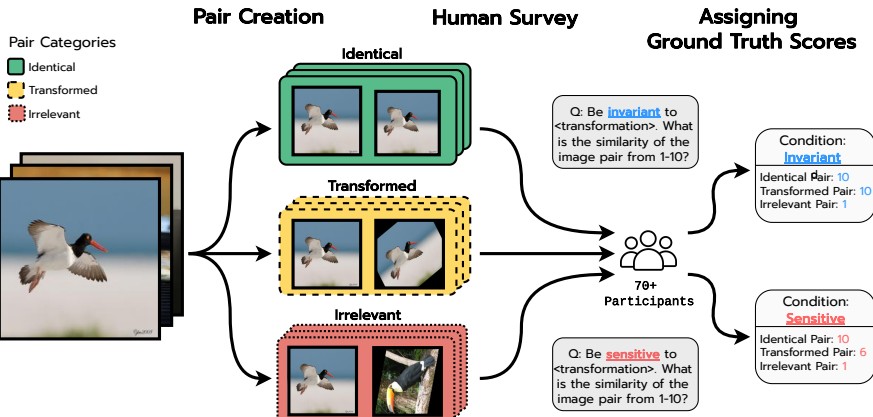

Figure 2: Dataset creation pipeline for the image-image datasets, i.e., PB$_{\text{COCO}}$, PB$_{\text{IN100}}$, and PB$_{\text{WU-II}}$. Each original image is used to create three pairs of image: identical, transformed, and irrelevant. Finally, based on the human study, each pair is scores depending on the condition of the prompt.

## 2 PAIRBENCH

### 2.1 DATASET CREATION

To evaluate how well vision-language models can assess similarity under controlled transformations, we construct a dataset using the PAIRBENCH framework, illustrated in Figure 2. For each original image, we generate three types of pairs: (1) Identical pairs where the second image is a near-duplicate, (2) Transformed pairs where a specific transformation (e.g., color jitter or spatial shift) is applied, and (3) Irrelevant pairs with unrelated content. We then gather similarity judgments from over 70 human annotators under two distinct conditions: invariant, where models should ignore transformations and focus on semantic similarity, and sensitive, where models should penalize such transformations. These human ratings verify the assignment of ground-truth similarity scores: both identical and irrelevant pairs are assigned fixed values of 10 and 1, respectively, while transformed pairs receive a score of 10 under invariance and 6 under sensitivity.

The framework is instantiated across image-only datasets (COCO, IN100) and image-text datasets (WhatsUp), using five standard image transformations plus a spatial position shift known to challenge VLMs. Full construction details, including transformation splits, prompt templates used to reduce linguistic bias, and details of the human study procedure, are provided in Appendix D.

### 2.2 METRICS

To measure the reliability of VLMs in scoring data pairs, we define four metrics that we measure across datasets and models: *MMScore*, $\varepsilon$-RelaxSym, Smoothness (*SM*), and Controllability (*Cont*).

We adopt the following notation to formulate the metrics: we denote the VLM being evaluated as $\mathcal{M}$ and the condition, which determines if the prompt instructs the model to be sensitive or invariant to a visual feature, as $C \in \{\texttt{sens}, \texttt{inv}\}$. Finally, given a dataset $\mathcal{D}_N = \{(d_1, d_2), (d_3, d_4), \ldots, (d_{2N-1}, d_{2N})\}$, we denote the similarity score of a data pair $(d_i, d_j) \in \mathcal{D}_N$ returned by an VLM ($\mathcal{M}$) for a given condition ($C$) as:

$$s_{\mathcal{M}}^C(d_i, d_j) := \mathcal{M}(C, d_i, d_j),$$

where $(d_i, d_j)$ could be an image-image or image-text pair. Note that we instruct the model to generate the output in a structured format to make sure the predicted score is parsable from the model output. If $s_{\mathcal{M}}^C(d_i, d_j)$ is valid, it would fall in the set $\mathcal{V} = [1, 10]$. However, models often do not consistently follow the details of the prompt and may produce scores not in $\mathcal{V}$ or outputs that do not satisfy the output format, in which case we set $s_{\mathcal{M}}^C(d_i, d_j) = -1$. Finally, to evaluate a model $\mathcal{M}$ on $\mathcal{D}_N$ given condition $C$, we create and annotate the set of all its outputs as:

$$S_{\mathcal{M}}^C(\mathcal{D}_N) = \left\{s_{\mathcal{M}}^C(d_i, d_j) \,\middle|\, (d_i, d_j) \in \mathcal{D}_N \cup \text{rev}(\mathcal{D}_N)\right\},$$

where $\text{rev}(\mathcal{D}_N) = \{(d_2, d_1), (d_4, d_3), \ldots, (d_{2N}, d_{2N-1})\}$ are the data pairs in reverse order.

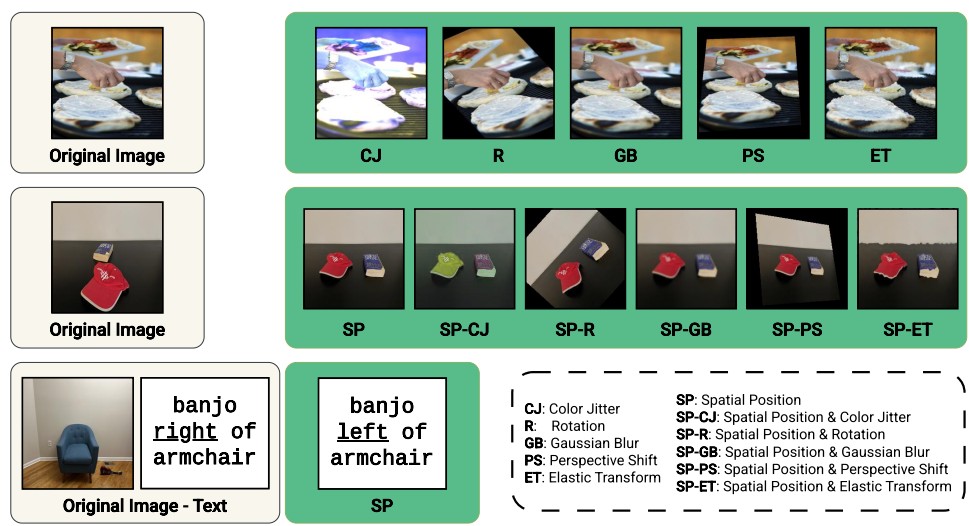

Figure 3: Examples of transformations (green boxes) applied to the original data points (gray boxes) of each subset instantiated with PAIRBENCH. The first row shows the different splits of PB$_{\text{COCO}}$ and PB$_{\text{IN100}}$, the second row for PB$_{\text{WU-II}}$, and the third for PB$_{\text{WU-IT}}$.

### 2.2.1 MMSCORE

We first introduce *MMScore*, the main metric of PAIRBENCH, which measures the alignment between model predictions and scores derived from human assessments. To this aim, we utilize Kendall's rank correlation coefficient (Kendall, 1938) between the predicted and the ground-truth scores. Instead of accuracy or squared error, we consider *MMScore* as we do not prompt the VLM with examples of the correct scores and hence cannot expect it to predict them directly. In other words, *MMScore* focuses on whether the VLM's scores are consistent with the ranking of the ground-truth scores without penalizing outputs that do not exactly match in magnitude. The better a model preserves the relative ordering and variance in the ground-truth scores, the better it is able to capture that characteristic. Hence we write,

$$MMScore(\mathcal{M}, C, \mathcal{D}_N) = \text{KT}(S_{\mathcal{M}}^C(\mathcal{D}_N), GT_C(\mathcal{D}_N)),$$

where $\text{KT}(.,.)$ is the Kendall's Tau and $GT_C(.)$ is the ground truth of the input dataset considering the condition of $C$. We further explore other rank-based metrics in Appendix C.2 and observed Kendall's Tau is the most suitable for this metric.

### 2.2.2 $\varepsilon$-RELAXSYM

The second metric we introduce aims to evaluate how consistent models are with respect to input order. This metric captures a fundamental characteristic when VLMs are used as re-rankers or automatic evaluators. Surprisingly, however, we found that most models do not satisfy exact symmetry, i.e., the equality of $sim(a, b)$ and $sim(b, a)$. We thus introduce $\varepsilon$-RelaxSym, which tolerates a difference of $\varepsilon$ between the scores that should be equal. More specifically, to analyze the symmetry of VLMs on a dataset $\mathcal{D}_N$, we compute the $\varepsilon$-RelaxSym of $(M)$ on $\mathcal{D}_{\mathcal{N}}$:

$$\varepsilon\text{-RelaxSym}(\mathcal{M}, \mathcal{D}_N) = \frac{1}{N} \sum_{(d_i, d_j) \in \mathcal{D}_N} \text{SoftEq}_\varepsilon(\mathcal{M}, d_i, d_j),$$

where $\text{SoftEq}_\varepsilon(\mathcal{M}, d_i, d_j)$ is defined as:

$$\text{SoftEq}_\varepsilon(\mathcal{M}, d_i, d_j) = \begin{cases} \mathbb{1}(|s_{\mathcal{M}}^C(d_i, d_j) - s_{\mathcal{M}}^C(d_j, d_i) \leq \varepsilon|), & s_{\mathcal{M}}^C(d_i, d_j), s_{\mathcal{M}}^C(d_j, d_i) \in \mathcal{V}, \\ 0, & \text{otherwise.} \end{cases}$$

Throughout this paper, we set $\varepsilon = 1$ and provide ablation studies covering other cases in Figure 9 in the Appendix.

### 2.2.3 SMOOTHNESS

We aim to measure how smooth the kernels induced by VLMs are. For instance, a non-smooth kernel would assign scores such that pairs are either exactly the same or completely different, while a smoother kernel produces more nuanced distinctions. We measure smoothness via the diversity of the predicted scores. Given $S_{\mathcal{M}}^{C}$, smoothness ($SM$) is computed as:

$$SM(\mathcal{M}, \mathcal{D}_N, C) = Ent(\{s \mid s \in S_{\mathcal{M}}^{C}(\mathcal{D}_N) \text{ and } s \in \mathcal{V}\}),$$

where $Ent(.)$ is the entropy of a set relative to its support, i.e., the set of candidate inputs.

### 2.2.4 CONTROLLABILITY

We measure how **responsive to instructions** models are. To do so, we define controllability based on the difference in *MMScore* between the sensitive and invariant settings. The more controllable a model is, the smaller the discrepancy observed between the `sens` and `invar` settings. Hence, when measuring the controllability on $\mathcal{D}_N$ for a model $\mathcal{M}$ is defined as

$$Cont(\mathcal{M}, \mathcal{D}_N) = 1 - \frac{|MMScore(\mathcal{M}, \texttt{sens}, \mathcal{D}_N) - MMScore(\mathcal{M}, \texttt{inv}, \mathcal{D}_N)|}{\sqrt{(MMScore(\mathcal{M}, \texttt{sens}, \mathcal{D}_N) \times MMScore(\mathcal{M}, \texttt{inv}, \mathcal{D}_N))}}.$$

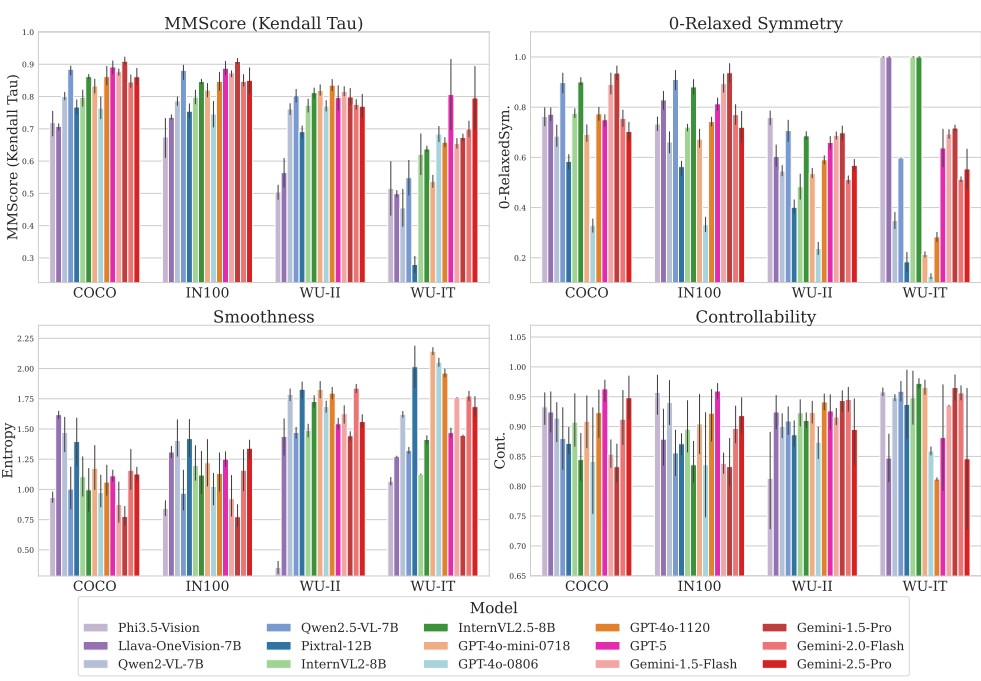

Figure 4: Best models performances on PB$_{\text{COCO}}$, PB$_{\text{IN100}}$, PB$_{\text{WU-II}}$, and PB$_{\text{WU-IT}}$. No model dominates the others as a similarity kernel, hence showing the limitation of defaulting to a single model as a judge for every task and dataset. Note the full symmetry of `Phi-3.5-vision`, `LLaVA-OneVision-7B`, and InternVL models on PB$_{\text{WU-IT}}$ are due to the lack of flexibility in the prompt structure to take the image anywhere but the beginning.

## 3 EVALUATION RESULTS

### 3.1 EXPERIMENTAL SETTING

We choose a comprehensive set of open- and closed-source vision-language models and evaluate them using the instantiations of PAIRBENCH. From the openly available models, we evaluated `Chameleon-7B` (Lu et al., 2024), `LLaVA-OneVision-7B` (Li et al., 2024), `Pixtral-12B` (Agrawal et al., 2024), `Phi-3.5-vision` (Abdin et al., 2024), four model sizes (1B, 2B, 4B,

and 8B) of InternVL2 (Wang et al., 2024c), four model sizes (1B, 2B, 4B, and 8B) of InternVL2.5 (Chen et al., 2024c), two capacities (2B and 7B) of Qwen2-VL (Wang et al., 2024b), two capacities (3B and 7B) Qwen2.5-VL (Hui et al., 2024), and three versions (`MolmoE-1B`, `Molmo-7B-O`, and `Molmo-7B-D`) of Molmo (Deitke et al., 2024).

We also considered commercial grade models and benchmarked four versions of GPT (Achiam et al., 2023)(`GPT-5`, `GPT-4o-0513`, `GPT-4o-0806`, `GPT-4o-1120`), `GPT-4o-mini-0718`, and four versions of Gemini (Reid et al., 2024) (`Gemini-1.5-Flash`, `Gemini-1.5-Pro`, `Gemini-2.0-Flash`, `Gemini-2.5-Pro`). Note that we consider multiple versions of the same architecture, as opposed to using the newest/largest version, to understand better how model capacity affects each of the metrics. We provide an extended analysis of different model versions in Appendix C.3. We run all open-source models on a single NVIDIA H100 GPU using greedy sampling for inference. For closed-source models, we use API access through either OPENROUTER[2] or OpenAI[3], applying the default inference hyperparameters provided by the respective platforms.

Also note that, since PAIRBENCH aims to evaluate VLMs as similarity kernels on image-only or text-image pairs, we do not evaluate text-only reasoning models such as OpenAI-o1 or DeepSeek-R1 (Guo et al., 2025). Further, we do not evaluate `Llama3.2-11B` (Grattafiori et al., 2024) as its official implementation on HuggingFace[4] does not support Flash Attention (Dao et al., 2022) and inference was prohibitively slow. As a result, we excluded them from our final results.

Table 1: Aggregated *MMScore*, *1-RS*:1-RelaxSym, *SM*, and *Cont* over all four data splits. No model performs the best across all metrics, showing the importance of PAIRBENCH to rank models based on different abilities.

| Model | MMScore(%) | 1-RS(%) | SM | Cont(%) |
|---|---|---|---|---|
| Phi-3.5-vision | 62.45 | 75.07 | 1.44 | 90.86 |
| Qwen2-VL-7B | 74.94 | 84.45 | 1.63 | 91.56 |
| Qwen2.5-VL-7B | 81.51 | 90.77 | 1.26 | 89.62 |
| InternVL2-8B | 77.07 | 74.63 | 1.32 | 91.62 |
| InternVL2.5-8B | 81.50 | 95.21 | 1.42 | 88.63 |
| Pixtral-12B | 68.61 | 74.84 | 1.67 | 88.44 |
| GPT-4o-1120 | 82.86 | 91.50 | 1.52 | 92.28 |
| GPT-5 | 83.63 | 96.51 | 1.39 | 93.71 |
| Gemini-1.5-Pro | 83.44 | 88.72 | 1.17 | 89.93 |
| Gemini-2.5-Pro | 80.80 | 83.67 | 1.44 | 90.7 |

Table 2: Spearman correlation of different metrics of PAIRBENCH with performance on other benchmarks for 23 models. *MMScore* has the highest correlation, making it the main metric.

| Metric | MMScore | 1-RS | SM | Cont |
|---|---|---|---|---|
| AI2D | 79% | 30% | 26% | 77% |
| HallusionBench | 80% | **36%** | 32% | 67% |
| MMBench | 77% | 29% | 33% | 71% |
| MMMU | **90%** | 34% | 31% | **80%** |
| MMStar | 81% | 20% | 33% | 79% |
| MMVet | 81% | 28% | **38%** | 68% |
| MathVista | 73% | 18% | 35% | 74% |
| OCRBench | 51% | 8% | 37% | 52% |

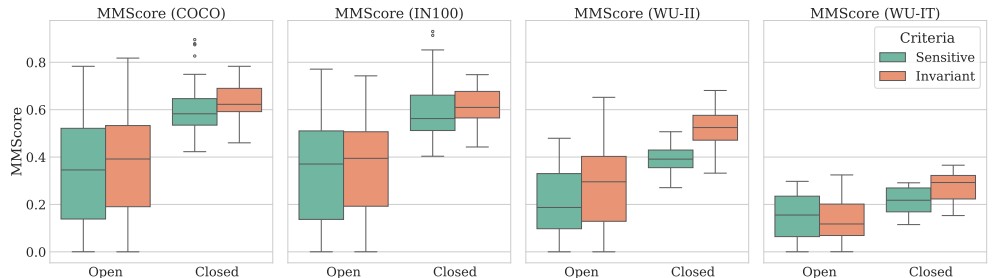

Figure 5: Closed- and open-source models perform comparable on image-text tasks. From the left to the right, the first three plots are image-image tasks, while the last is an image-text comparison task.

## 3.2 RESULTS

We analyze and plot the results of the best models in Figure 4 and provide an aggregated version of the metrics over all four datasets in Table 1. We aggregate different splits/datasets by taking the average

---

[2]https://openrouter.ai/

[3]https://platform.openai.com/

[4]https://huggingface.co/

of them to give each sub-dataset equal importance in the final number. The full set of benchmarking results of all models for PAIRBENCH on all datasets and metrics are reported in Appendix C.

### 3.2.1 GENERAL OBSERVATIONS

As illustrated in Figure 4 and Table 1, we observe no model, whether closed- or open-source, is the best performer across all four metrics. Moreover, we further observe that for each metric, no model is the 'best' similarity kernel across the four different datasets either. This shows how features of the dataset and also the metrics a user might want to optimize play a crucial role in which VLM to choose as the best similarity kernel/judge. For instance, among open-source models, although `InternVL2.5-8B` outperforms the rest in *MMScore*, it is less controllable and smooth than `Qwen2-VL-7B` or `LLaVA-OneVision-7B`.

When considering PAIRBENCH's main metric, *MMScore*, we notice that the performance of models is generally better on image-image pairs rather than image-text pairs. Furthermore, as seen in Figure 5, we observe that although open-source VLMs are roughly comparable to closed-source ones on $PB_{WU\text{-}IT}$, the gap between the two groups is larger in the image-image pairs. However, `InternVL2.5-8B` is a strong competitor to closed-source models considering all four metrics and could potentially be used as a substitute to closed-source models as a similarity kernel based on the results reported in Table 1.

Interestingly, we further observe a pattern regarding `GPT-4o-1120`, a common default judge used in the literature, and its lower cost version, `GPT-4o-mini-0718`; they both suffer from low 1-RelaxSym when comparing image-text pairs, and the cheaper model's *Cont* and *SM* are higher or comparable to that of the expensive one across datasets. Another fascinating result we observed was the effect the scaling effect on different metrics of PAIRBENCH for a single model family; the larger a model gets, the better it performs on *MMScore* and 1-RelaxSym. However, that does not hold for controllability and smoothness. This emphasizes the importance of PAIRBENCH in analyzing the capabilities of models, both open and closed-source, as similarity kernels to be better used as judges. We analyze and plot these results further in Appendix C and show further qualitative examples of the errors the best VLMs make in these tasks in Appendix A.

### 3.2.2 CORRELATION WITH BENCHMARKS

To showcase the effectiveness of our metrics and PAIRBENCH in predicting reasoning performance, we compute the Spearman correlation with respect to other popular benchmarks used in the literature. By showing correlations of our metrics with these benchmarks, we show that although the PAIR-BENCH framework introduces simple and cost-efficient methods focused on evaluating the ability to compare due to prompted VLMs, these metrics are predictive of an VLM's performance on other tasks, and can provide an alternative for model ranking and validation during development.

We collect all the model performances from the OPENVLM LEADERBOARD(Duan et al., 2024) and filter out the models we evaluate, resulting in all 27 (including different versions/capacities of closed- and open-source) models. By filtering out the benchmarks that have evaluation scores for all 27 models on OpenVLM, we end up with AI2D (Kembhavi et al., 2016), HallusionBench (Guan et al., 2024), MMBench (Liu et al., 2025), MMStar (Chen et al., 2024b), MMMU (Yue et al., 2024), MathVista (Lu et al., 2023), MM-Vet (Yu et al., 2023), OCRBench (Liu et al., 2024b). Each metric is aggregated for each model across all the configurations created by PAIRBENCHbefore computing correlations. Namely, we aggregate all features within each dataset (e.g., CJ, SP, etc.) and across all datasets (e.g., $PB_{COCO}$, $PB_{WU\text{-}II}$) and end up with an aggregate result per metric for each model.

As seen in Table 2, all metrics in PAIRBENCH have a high positive correlation with performances in benchmarks. More specifically, we observe that *MMScore* has strong correlations with all benchmarks, indicating that it aligns closely with overall model performance. Hence, we select it as the main metric of PAIRBENCH. Furthermore, when analyzing the correlations of PAIRBENCH's other metrics with all benchmarks, we find that the strength of correlation reflects how much of the base skill captured by the metric is required by each benchmark. For example, HallusionBench shows the highest correlation with 1-RelaxSym, which is notable since HallusionBench focuses primarily on evaluating hallucinations in VLMs. This suggests a connection between lack of symmetry and hallucination. Another example is the highest and lowest correlation of *Cont* with MMMU and OCRBench, respectively. MMMU's prompt mostly contain complex questions and multiple answers, whereas OCRBench features simple prompts for most questions. We hypothesize that since *Cont*

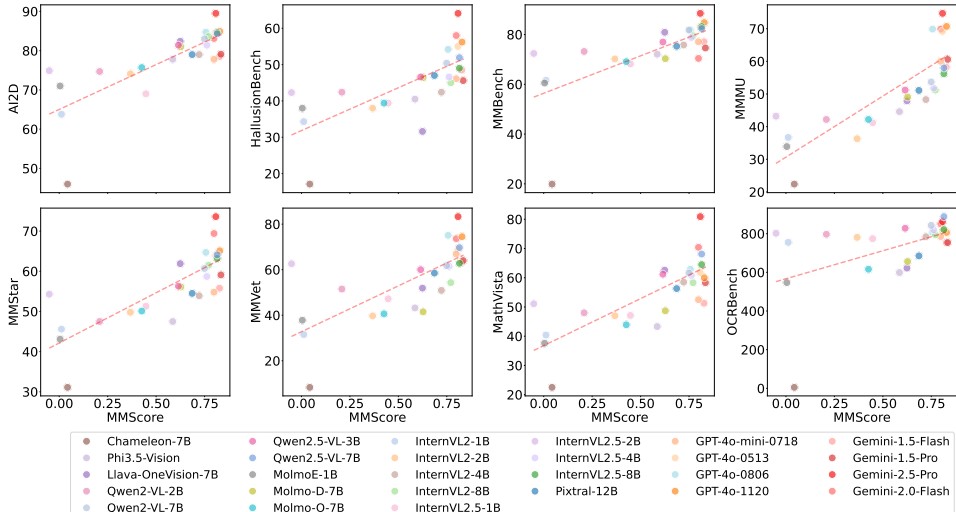

Figure 6: The main metric of PAIRBENCH, *MMScore*, strongly correlates with previous multimodal benchmarks, showcasing its predictive power of a model's performance at a lower cost to create.

measures how well models follow the prompt, these differences explain the highest and lowest correlations observed with MMMU and OCRBench.

Note that measuring comparison skills incurs a low cost as it does not require expert-generated annotations. Our results suggest that metrics that assess these skills can serve as a low-cost surrogate of performance in various tasks: an efficient alternative to model selection. We further show scatter plots that highlight correlations in Figure 6, and more comprehensively in Figure 12 in Appendix C.

### 3.2.3 PROMPT SELECTION

We show the *MMScore* performance of models across datasets in Figure 7 for various prompt templates. As evident from the plots, no single prompt template consistently achieves the best performance across all models. Some models perform better with certain phrasings, while others are negatively affected by the same templates. This variation highlights the significant influence of prompt wording on model behavior. Recent work (Polo et al., 2024b) has emphasized the importance of using diverse prompts. Similarly, our randomized approach offers empirical support for that recommendation. Evaluating models with multiple prompt templates and averaging the results eliminates prompt-induced variance and leads to more reliable and fair comparisons. We recommend this strategy as a stronger and more principled standard for future benchmarking of prompted models, whether multimodal or otherwise.

## 4 RELATED WORK

Recent work has explored using language models as automated evaluators in NLP and vision-language domains, with approaches like GPTSCORE and G-eval (Fu et al., 2023; Liu et al., 2023) showing alignment with human preferences. However, concerns remain regarding their reliability, especially due to known limitations such as sensitivity to input order (Fang et al., 2024) and failure to infer reversible relationships (Berglund et al., 2023). In the multimodal case, work such as Zheng et al. (2023); Thakur et al. (2024); Murugadoss et al. (2024) evaluates VLMs as judges, highlighting issues of bias, prompt dependency, and limited control over evaluation criteria. Our work extends this line of research by focusing on structured pairwise comparisons, measuring not just performance alignment but also properties like symmetry, smoothness, and controllability.

While benchmarks like Chen et al. (2024a) and Awal et al. (2024) introduce ways to test comparison abilities of VLMs, we aim to provide a more systematic and transformation-aware framework. Prior work also identifies well-known blind spots in discriminative models such as CLIP, including spatial reasoning failures (Kamath et al., 2023) and neglect of logical constructs like negation (Alhamoud et al., 2025). Our goal is to support the development and evaluation of models in these areas through

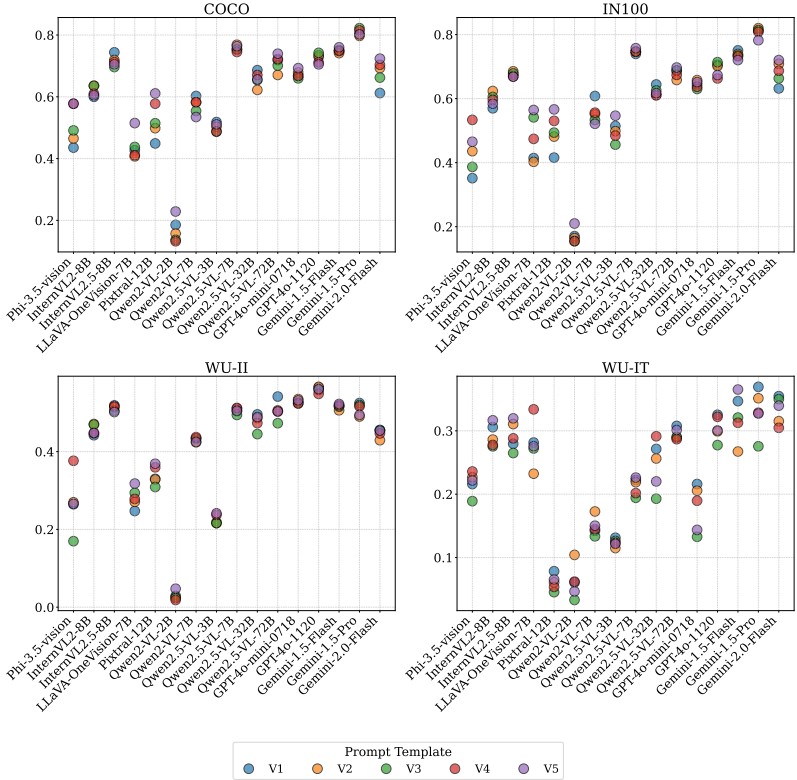

Figure 7: By using multiple prompt templates, we ensure no model is biased towards a single prompt and the mean capture the overall model performance. Above we show *MMScore*.

a carefully designed testbed. A detailed review of related evaluation benchmarks, model limitations, and pair-comparison studies is included in Appendix B.

## 5 CONCLUSION AND FUTURE WORK

We introduced PAIRBENCH, a framework that systematically evaluates the fundamental yet over-looked ability of VLMs to compare visual inputs, a capability critical for applications ranging from automated evaluation and re-ranking to retrieval-augmented generation. By focusing on four key metrics (alignment with human judgment, symmetry, smoothness, and controllability), PAIRBENCH provides comprehensive insights into how models process comparative information while intention-ally minimizing computational requirements. This cost-efficiency addresses growing concerns around the unsustainable costs of model evaluation, which increasingly constitutes a significant portion of model development budgets (Polo et al., 2024a; Pacchiardi et al., 2024; Yuan et al., 2025).

Our extensive benchmarking revealed that no model excels across all metrics, with even leading commercial systems demonstrating concerning asymmetries when comparing identical pairs in different orders. Particularly noteworthy is our finding that performance on PAIRBENCH strongly correlates with results on complex reasoning benchmarks, suggesting that *comparison capabilities may constitute a fundamental skill* that underlies performance across diverse tasks. This insight offers a more efficient path to model selection and validation without the computational burden of exhaustive evaluations on large-scale benchmarks. As a means to further improve evaluation efficiency while accounting for sensitivity to prompting, we applied a randomized prompting strategy, rendering comparisons across models more reliable at no additional inference cost.

Looking forward, we hypothesize that tailored post-training approaches to focus on improving comparative skills and better model classes may enhance overall capabilities across diverse tasks, given the transferability our results revealed. Future research could explore architectural modifications or specialized fine-tuning techniques that optimize for these metrics, creating more reliable VLMs for critical evaluation tasks.

**Ethics Statement**   We do not foresee any ethical issues with our work and have complied with the conference guidelines. We believe our work will aid in the better and safer usage of VLMs in practice, as we offer means to evaluate and understand model behavior prior to deployment. This can help mitigate potential risks associated with model biases and inconsistencies, leading to more reliable and transparent systems. As per language model usage, besides the evaluations we reported, we restricted ourselves to using these only to refine or rephrase handwritten parts of the manuscript to ensure correctness and clarity.

**Reproducibility Statement**   We disclose all the details needed to carry out our evaluation such as exact prompt templates, exact model versions, and inference settings.

A sample of the data and example code to compute metrics are included as part of the supplementary material, and the full dataset is made anonymously available at `https://anonymous.4open.science/r/pairbench-6C08`.

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

# Table of Contents – Appendix

# A  ERROR ANALYSIS

In this section, we look into the outputs of each model and their errors given different data pairs. More specifically, we look into errors made by `Gemini-1.5-Pro`, `GPT-4o-1120`, `InternVL2.5-8B`, `Pixtral-12B`, `Qwen2-VL-7B`, and `Phi-3.5-vision`.

## A.1  GEMINI MODEL

Below is an error example of `Gemini-1.5-Pro` on a data-pair from PB$_{\text{COCO}}$with color jittering (CJ).

---

**`Gemini-1.5-Pro` Evaluation Instructions**

**User prompt**: Evaluate the similarity of the images based on the following conditions: - *Be invariant to color jittering while evaluating images. Even if one image has been modified with random color changes (e.g., brightness or contrast adjustments), the similarity score should remain high. If the images are different, the similarity score should decrease regardless of the jittering.*

The score should take into account how well the images align with these conditions, as well as their overall resemblance. Even if the images are identical but fail to meet the conditions, they should still receive a higher score than completely different images. Provide a score from 1 to 10, where 1 indicates no similarity and 10 indicates identical images that fully satisfy the conditions. Respond only in this format:

```
Score: <1-10>
Reason: <reason for score>
```

Nothing else should be included. What score would you give? "`data1`" "`data2`"

---

**`Gemini-1.5-Pro` Response - Original Order**  **`Gemini-1.5-Pro` Response - Reversed Order**

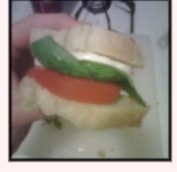 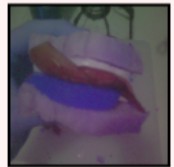       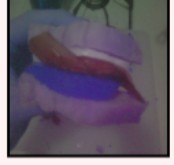 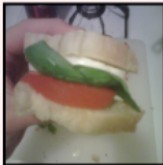

**Model response:**

> *Score: 10*
> *Reason: The images are of the same sandwich, one in normal color and the other with a strong purple color filter applied. Despite the heavy color distortion, the content is identical, fulfilling the color invariance requirement.*

**Model response:**

> *Score: 1*
> *Reason: The first image appears to be a mold of some sort, possibly for dental work or other medical purposes. The second image is a Caprese sandwich. They are entirely different objects. While the layered structure might be vaguely similar, the content, context, and purpose are completely dissimilar.*

## A.2 GPT4O MODEL

Below is an error example of `GPT-4o-1120` on a data-pair from $PB_{COCO}$ with color jittering (CJ).

---

**GPT-4o-1120 Evaluation Instructions**

**User prompt**: Evaluate the similarity of the images based on the following conditions:
- *Be invariant to color jittering while evaluating images. Even if one image has been modified with random color changes (e.g., brightness or contrast adjustments), the similarity score should remain high. If the images are different, the similarity score should decrease regardless of the jittering.*
The score should take into account how well the images align with these conditions, as well as their overall resemblance. Even if the images are identical but fail to meet the conditions, they should still receive a higher score than completely different images.
Provide a score from 1 to 10, where 1 indicates no similarity and 10 indicates identical images that fully satisfy the conditions.
Respond only in this format:

```
Score: <1-10>
Reason: <reason for score>
```

Nothing else should be included. What score would you give? `"data1"` `"data2"`

---

**GPT-4o-1120 Response - Original Order**

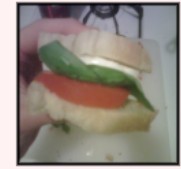 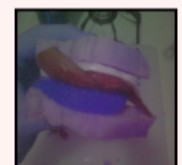

**Model response:**

*Score: 8*
*Reason: The images are of the same sandwich, with similar shapes and arrangements of the bread, tomato, mozzarella, and basil. The second image has substantial color jittering, but since the structural alignment and composition remain consistent, the similarity remains high.*

**GPT-4o-1120 Response - Reversed Order**

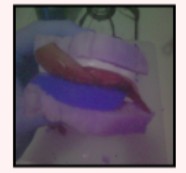 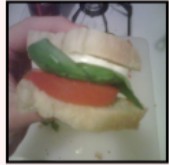

**Model response:**

*Score: 3*
*Reason: The images depict similar objects, presumably a sandwich with layers, but there are notable differences. The first image appears distorted with altered coloration, and while it shares structural similarities with the second image, the overall resemblance is weak.*

---

## A.3 INTERNVL2.5 MODEL

Below is an error example of `InternVL2.5-8B` on a data-pair from $PB_{COCO}$ with that are irrelevant to each other.

---

**InternVL2.5-8B Evaluation Instructions**

**User prompt**: Evaluate the similarity of the images based on the following conditions: - *Be invariant to Gaussian blurs while evaluating images. For example, even if both images are the exact same, even if one is more blurred compared to the other, the similarity score should remain high. If the images are different, the similarity score should decrease regardless of how blurred they are.* The score should take into account how well the images align with these conditions, as well as their overall resemblance. Even if the images are identical but fail to meet the conditions, they should still receive a higher score than completely different images.

Provide a score from 1 to 10, where 1 indicates no similarity and 10 indicates identical images that fully satisfy the conditions. Respond only in this format:

```
Score: <1-10>
Reason: <reason for score>
```

Nothing else should be included. What score would you give? "data1" "data2"

---

**InternVL2.5-8B - Original Order**

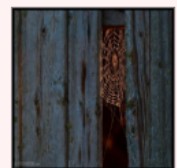 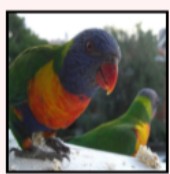

**Model response:**

> *Score: 7*
> *Reason: The images are similar in content, featuring parrots, but they are not identical. The first image is a close-up of a spider web on a wooden surface, while the second image shows two parrots on a ledge. The parrots are the main subjects in the second image, and the first image has a different focus and subject matter. The images do not meet the condition of being identical, but they share some visual elements like color and subject type.*

**InternVL2.5-8B - Reverse Order**

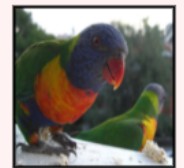 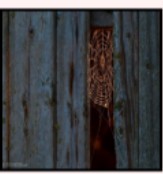

**Model response:**

> *Score: 1*
> *Reason: The images are completely different in terms of content, subject matter, and context. The first image features colorful parrots, while the second image shows a wooden door with a spider web. There is no similarity in terms of subject, color, or composition.*

---

A.4    PIXTRAL MODEL

Below is an error example of `Pixtral-12B` on a data-pair from PB$_{COCO}$ with rotation (R).

---

**Pixtral-12B Evaluation Instructions**

**User prompt**: Rate the similarity of these images on a scale of 1 to 10 while considering the following condition:

*- Be invariant to rotations while evaluating images. For example, if one image is a rotated version of the other, the similarity score should remain high. If the images are different, the similarity score should decrease regardless of their orientation.*

The score should account for both how well the images satisfy the conditions and their overall resemblance. If the images are identical but do not meet the conditions, they should still score higher than unrelated images.

Your response must follow this exact format:

```
Score: <1-10>
Reason: <reason for score>
```

Provide nothing else. What's your score? "data1" "data2"

---

**Pixtral-12B Response - Original Order**

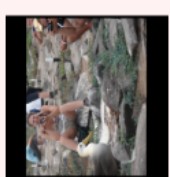 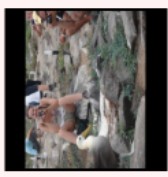

**Model response:**

*Score: 8*
*Reason: The images depict similar scenes of people interacting with birds in natural settings. The first image features a group of people sitting on rocks with birds nearby, while the second image has a person extending their hand towards a bird in a mountainous area. Despite slight differences in the environment and specific actions, the overall theme and presence of birds and human interaction score high.*

**Pixtral-12B Response - Reversed Order**

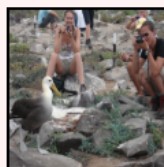

**Model response:**

*Score: 2*
*Reason: The images depict completely different scenes and subjects, with no recognizable similarity between them. One image shows a bird and people in a rocky area, while the other does not bear any resemblance to it.*

## A.5 QWEN2-VL MODEL

Below is an error example of `Qwen2-VL-7B` on a data-pair from PB$_{COCO}$ with gaussian blur (GB).

---

**Qwen2-VL-7B Evaluation Instructions**

**User prompt**: As a similarity evaluator, your responsibility is to assess the similarity of the given images while considering these conditions: - *Be sensitive to Gaussian blurs while evaluating images. For example, if both images are the same except one is a blurred version of the other, the similarity score should decrease. If the images are the same and have the same exact sharpness, the similarity score should remain high.*

The similarity score should reflect both how well the images align with the conditions and their overall resemblance. Images that are identical but do not meet the conditions should receive a moderate score, while completely unrelated images should receive the lowest score. Provide your score on a scale of 1 to 10, with 10 being identical images that fully meet the conditions. Ensure your response is in the following format:

```
Score: <1-10>
Reason: <reason for score>
```

Provide nothing else. What is your score? "data1" "data2"

---

**Qwen2-VL-7B - Original Order**

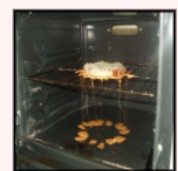 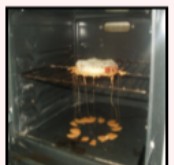

**Model response:**

> *Score: 6*
> *Reason: The images are very similar, with the second image being a slightly blurred version of the first. However, they do not fully meet the conditions as the images are not identical and the blurring is not perfect.*

**Qwen2-VL-7B - Reverse Order**

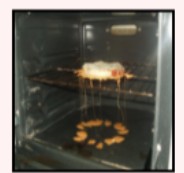 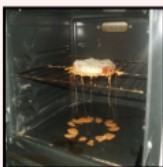

**Model response:**

> *Score: 9*
> *Reason: The images are nearly identical, with only a slight difference in the sharpness of the objects. The overall resemblance is high, and the conditions are met.*

---

## A.6  PHI3.5 MODEL

Below is an error example of `Phi-3.5-vision` on a data-pair from PB$_{COCO}$ with elastic transform (ET).

---

**`Phi-3.5-vision` Evaluation Instructions**

**User prompt**: Judge the similarity of these images based on: - *Be sensitive to elastic transformations while evaluating images. For example, if the images are the same but have differing distortions due to elastic transformations, the similarity score should decrease. If the images are identical and have matching elastic transformations, the similarity score should remain high.*

The similarity score should reflect both the overall resemblance of the images and how well they satisfy the conditions. Identical images that do not meet the conditions should still score higher than completely unrelated images.

Provide a score on a scale of 1 to 10, with 1 being no similarity and 10 being identical images that satisfy all conditions. Respond strictly in this format:

```
Score: <1-10>
Reason: <reason for score>
```

Do not include additional text. What's your rating? "`data1`" "`data2`"

---

**`Phi-3.5-vision` - Original Order**

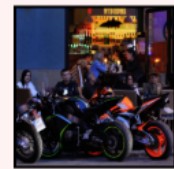 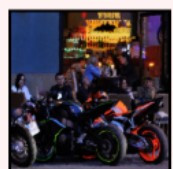

**Model response:**

> *Score: 1*
> *Reason: The images are highly distorted with elastic transformations, making it impossible to accurately compare the content.*

**`Phi-3.5-vision` - Reverse Order**

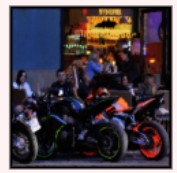 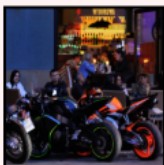

**Model response:**

> *Score: 10*
> *Reason: The images are identical with no differences in content, color, or composition. They both depict the same scene with motorcycles and people in a bar setting, and the elastic transformations do not alter the image in any way.*

## B   EXTENDED RELATED WORK

Using language models as automatic evaluators has become a somewhat common practice with popular approaches such as GPTSCORE and G-eval (Fu et al., 2023; Liu et al., 2023) being used to rank responses in the NLP domain. Due to that there has been a significant amount of recent work that has investigated the capabilities and limitations of using LLMs as judges (Thakur et al., 2024; Chiang & Lee, 2023; Murugadoss et al., 2024; Shankar et al., 2024). Chiang & Lee (2023) have shown that LLM evaluations are consistent and reproducible, making them suitable alternatives for human evaluation, they argue that these models inherent biases should prevent them using independently rather than *alongside* human experts. Furthermore, Zheng et al. (2023) reveal that large VLMs, e.g., GPT-4 Turbo, align well with human judgments and Thakur et al. (2024) further states that simpler models may still outperform GPT-4 Turbo in ranking tasks due to superior alignment metrics. Also, recent work assessed how humans can help LLMs evaluate better by testing different instruction

types or designing tools that result in more balanced evaluations (Murugadoss et al., 2024; Shankar et al., 2024).

It is worth noting that known limitations of LLMs such as their lack of invariance to the order of examples given in a prompt, which is a well studied issue of natural language models (Fang et al., 2024), may render auto evaluation unreliable. Similarly, Berglund et al. (2023) show failure cases where models trained on unidirectional relationships do not infer the reverse, indicating systemic limitations even in state-of-the-art LLMs such as GPT-4 (as seen in Figure 1 and in Appendix A for VLMs). Our main goal is to investigate the reliability of automated evaluation in the multimodal context, by probing the models to compare data pairs.

Namely, the evaluations we carry out focus on testing in multiple different ways how good VLMs are when it comes to comparing data instances, such as whether VLMs prompted to compare are symmetric or smooth for instance, and to what extent they can be controlled, i.e., instructed to pay attention to or ignore certain features of the inputs. While the literature is more sparse regarding testing VLMs in this setting, recent work has tested for something along those lines. Chen et al. (2024a) for instance propose a benchmark for evaluating VLMs in multiple different scenarios, including checking whether pairwise comparisons of responses to a query correlated with human judgments. They concluded that although correlations are relatively high on comparison tasks, biases and inconsistencies affect performance on pair scoring and batch ranking. Similarly, Awal et al. (2024) introduced a synthetic dataset containing paired images that differ only along one feature (e.g., the color of an object). We seek to add to this branch of the literature by introducing a framework where controlled experiments can be carried out to anticipate the performance of models when being used as judges, and various different characteristics of automatic judges can be identified (e.g., how smooth they are).

Unlike the case of generative VLMs discussed above, discriminative visual language models such as CLIP (Radford et al., 2021) are covered by a significant amount of recent work, and several failure modes are well reported, mostly deriving from the fact this class of VLMs tends to behave as bag-of-words models, focusing on nouns and ignoring relationships and semantics in their input data (Yuksekgonul et al., 2023). For instance, CLIP was observed to struggle with spatial reasoning (Kamath et al., 2023) and ignore negation (Alhamoud et al., 2025). On the other hand, fine-tuning CLIP to reason about pairwise differences Sam et al. (2024) showed that discriminative VLMs can improve on how well they manage to reason about pairwise differences if training is tailored for enabling so, highlighting the benefits that being able to measure these skills may inform training and improve models as a consequence. Ouali et al. (2024) showed that fine-tuning generative VLMs to turn them into discriminative models results in improved image-retrieval from text, which aligns with results we reported in Section C.5 showing a gap between open-sources VLMs and CLIP-style encoders.

## C  FULL RESULTS

In this section, we provide the *MMScore* of all models on all the different splits of $PB_{COCO}$, $PB_{IN100}$, $PB_{WU-II}$, and $PB_{WU-IT}$ in Tables 3, 4, 3, 4, 5, 6, and 7. We further report the coverage, the number of times the VLMs give a valid output, of each model on our different proposed datasets.

### C.1  ALL $\varepsilon$-RELAXSYM FOR DIFFERENT $\epsilon$S

To show the $\varepsilon$-RelaxSym for different values of $\varepsilon$, we plot Figure 9 and show as $\varepsilon$ gets higher, the values go higher. However, some models such as the GPT4o models struggle with symmetry. Please note that if $\varepsilon = 0$, it is the same as not having a threshold and hence calculating exact symmetry rather than a relaxed version.

### C.2  VARIATIONS OF MMSCORE FORMULATION

In this section, we examine how model rankings vary when Kendall's Tau is replaced with other correlation-based metrics, namely Normalized Mutual Information (NMI), Spearman's rank correlation, and Normalized Discounted Cumulative Gain (NDCG). Overall, we find that NMI, Spearman, and NDCG produce rankings consistent with those obtained using Kendall's Tau. However, NDCG

Table 3: Comparison of the *MMScore* metric ($\times 100$) of VLMs on PB$_{\text{COCO}}$ and PB$_{\text{IN100}}$ benchmarks in the *sensitive* setting. Models are evaluated across multiple criteria: color jitter (CJ), elastic transform (ET), gaussian blur (GB), perspective shift (PS), and rotation (R). Higher scores indicate better performance.

| Model | PB$_{\text{COCO}}$ | | | | | PB$_{\text{IN100}}$ | | | | |
|---|---|---|---|---|---|---|---|---|---|---|
| | CJ | ET | GB | PS | R | CJ | ET | GB | PS | R |
| Chameleon-7B | 00.37 | 00.34 | 00.19 | 00.31 | 00.60 | 00.38 | 00.26 | 00.31 | 00.50 | 00.52 |
| LLaVA-OneVision-7B | 36.51 | 44.05 | 38.57 | 43.80 | 41.41 | 37.05 | 49.89 | 40.00 | 46.01 | 49.30 |
| Phi-3.5-vision | 38.21 | 51.61 | 61.94 | 47.33 | 34.56 | 25.74 | 43.03 | 51.40 | 32.51 | 23.61 |
| Pixtral-12B | 37.67 | 56.25 | 54.32 | 49.53 | 36.80 | 30.75 | 52.30 | 51.94 | 46.04 | 40.76 |
| InternVL2-1B | 03.23 | 03.47 | 03.27 | 03.63 | 03.51 | 02.59 | 02.38 | 01.70 | 02.02 | 02.23 |
| InternVL2-2B | 23.89 | 32.76 | 34.32 | 31.53 | 24.76 | 18.32 | 34.02 | 33.35 | 28.17 | 23.35 |
| InternVL2-4B | 52.13 | 69.43 | 62.46 | 63.77 | 52.68 | 45.25 | 65.90 | 59.90 | 60.28 | 51.04 |
| InternVL2-8B | 51.58 | 62.80 | 62.35 | 60.27 | 54.80 | 47.94 | 60.18 | 58.60 | 56.66 | 53.00 |
| InternVL2.5-1B | 16.74 | 25.38 | 27.67 | 24.83 | 16.54 | 15.63 | 33.67 | 39.23 | 37.97 | 22.53 |
| InternVL2.5-2B | 12.48 | 19.58 | 25.26 | 18.33 | 13.84 | 17.27 | 38.28 | 39.21 | 31.23 | 21.45 |
| InternVL2.5-4B | 42.61 | 59.78 | 54.33 | 55.34 | 49.47 | 41.35 | 62.35 | 54.21 | 56.18 | 49.90 |
| InternVL2.5-8B | 54.51 | 73.37 | 78.31 | 63.17 | 60.71 | 51.76 | 77.10 | 76.40 | 60.40 | 55.30 |
| MolmoE-1B | 00.40 | 00.09 | 01.20 | 00.03 | 00.05 | 00.41 | 00.01 | 00.45 | 00.01 | 00.01 |
| Molmo-7B-O | 14.32 | 16.02 | 48.93 | 16.12 | 15.40 | 12.91 | 14.20 | 48.43 | 13.83 | 12.16 |
| Molmo-7B-D | 27.06 | 45.28 | 34.46 | 49.60 | 30.39 | 22.88 | 41.06 | 35.83 | 44.49 | 32.22 |
| Qwen2-VL-2B | 09.91 | 11.82 | 09.01 | 13.13 | 11.95 | 10.63 | 13.69 | 10.41 | 13.21 | 12.23 |
| Qwen2-VL-7B | 42.58 | 61.90 | 50.22 | 55.81 | 51.10 | 38.24 | 61.73 | 50.23 | 53.07 | 52.29 |
| GPT-4o-mini-0718 | 49.98 | 65.97 | 58.29 | 53.23 | 53.60 | 47.06 | 67.06 | 56.43 | 49.97 | 52.59 |
| GPT-4o-0513 | 50.96 | 65.54 | 61.67 | 56.69 | 56.71 | 48.55 | 65.68 | 57.48 | 54.11 | 55.00 |
| GPT-4o-0806 | 42.26 | 60.58 | 56.62 | 50.13 | 53.63 | 40.35 | 60.66 | 52.65 | 49.62 | 49.77 |
| GPT-4o-1120 | 51.31 | 63.50 | 61.35 | 57.84 | 57.16 | 50.88 | 66.55 | 58.14 | 56.25 | 55.52 |
| Gemini-1.5-Flash | **58.26** | 82.64 | 87.41 | 65.92 | 61.08 | **56.25** | 79.69 | 85.21 | 62.07 | 61.15 |
| Gemini-1.5-Pro | 53.33 | **87.86** | **89.56** | **74.92** | **71.04** | 51.19 | **91.36** | **92.98** | **71.56** | **74.22** |

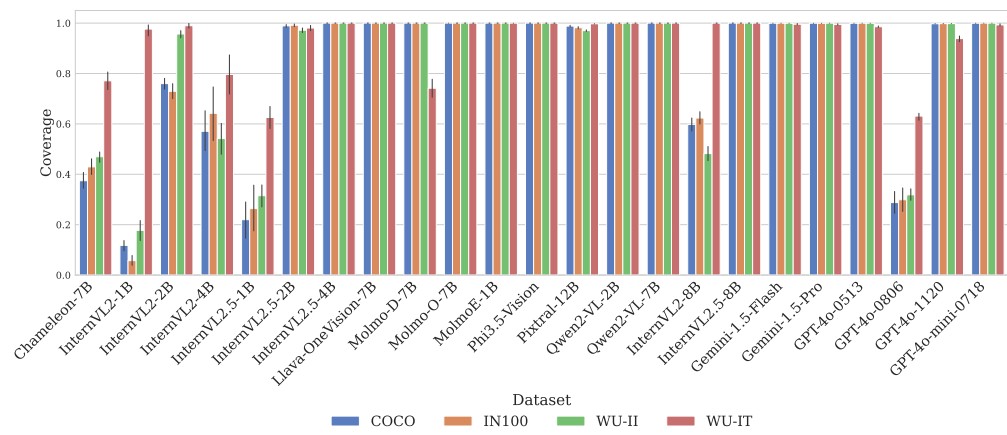

Figure 8: Coverage of each model.

exhibits lower sensitivity, leading to flatter curves that are less informative for our setting. We attribute this to the fact that NDCG was originally designed for retrieval systems, where the top-ranked items dominate the evaluation, and thus the metric emphasizes alignment at the highest scores rather than capturing fine-grained differences across the full ranking. The results can be seen in Fig. 10.

Table 4: Comparison of the *MMScore* metric ($\times 100$) of VLMs on $PB_{COCO}$ and $PB_{IN100}$ benchmarks in the *invariant* setting. Models are evaluated across multiple criteria: color jitter (CJ), elastic transform (ET), gaussian blur (GB), perspective shift (PS), and rotation (R). Higher scores indicate better performance.

| Model | $PB_{COCO}$ | | | | | $PB_{IN100}$ | | | | |
|---|---|---|---|---|---|---|---|---|---|---|
| | CJ | ET | GB | PS | R | CJ | ET | GB | PS | R |
| Chameleon-7B | 00.89 | 00.34 | 00.44 | 00.51 | 00.38 | 00.57 | 00.35 | 00.53 | 00.58 | 00.45 |
| LLaVA-OneVision-7B | 35.13 | 37.26 | 39.22 | 40.29 | 38.29 | 38.09 | 43.04 | 41.83 | 40.86 | 42.24 |
| Phi-3.5-vision | 49.41 | 40.19 | 42.93 | 55.03 | 47.90 | 45.88 | 33.79 | 39.72 | 50.41 | 39.46 |
| Pixtral-12B | 48.26 | 47.34 | 45.35 | 60.20 | 55.65 | 41.53 | 45.30 | 42.84 | 52.63 | 52.65 |
| InternVL2-1B | 02.69 | 01.76 | 02.71 | 02.00 | 02.69 | 01.39 | 00.82 | 01.22 | 00.90 | 01.40 |
| InternVL2-2B | 36.38 | 31.55 | 31.99 | 39.18 | 37.28 | 32.68 | 31.40 | 30.13 | 35.98 | 34.70 |
| InternVL2-4B | 59.44 | 55.47 | 51.35 | 59.61 | 59.02 | 51.74 | 52.77 | 49.60 | 54.63 | 53.11 |
| InternVL2-8B | 58.69 | 58.56 | 53.60 | 61.91 | 64.22 | 58.44 | 54.48 | 51.78 | 61.97 | 62.90 |
| InternVL2.5-1B | 21.39 | 18.59 | 21.65 | 23.19 | 22.86 | 22.52 | 14.63 | 24.34 | 22.76 | 19.24 |
| InternVL2.5-2B | 22.85 | 19.05 | 21.46 | 27.62 | 25.99 | 32.09 | 33.03 | 37.34 | 34.65 | 34.75 |
| InternVL2.5-4B | 56.24 | 47.41 | 43.93 | 53.71 | 55.28 | 61.80 | 50.50 | 47.33 | 51.58 | 58.56 |
| InternVL2.5-8B | **75.11** | 65.18 | 66.32 | **78.56** | **81.77** | **72.53** | 61.61 | 62.23 | 65.18 | 74.27 |
| MolmoE-1B | 00.10 | 00.11 | 00.06 | 00.02 | 00.00 | 00.02 | 00.11 | 00.10 | 00.07 | 00.25 |
| Molmo-7B-O | 26.86 | 34.58 | 33.46 | 34.70 | 24.55 | 25.04 | 30.81 | 38.52 | 32.79 | 27.65 |
| Molmo-7B-D | 47.20 | 45.02 | 43.02 | 50.54 | 48.64 | 45.01 | 45.83 | 45.47 | 49.25 | 40.87 |
| Qwen2-VL-2B | 09.55 | 09.10 | 10.21 | 12.65 | 08.83 | 09.02 | 09.61 | 10.01 | 14.97 | 09.33 |
| Qwen2-VL-7B | 50.52 | 51.80 | 52.70 | 54.50 | 53.29 | 47.86 | 49.73 | 51.18 | 51.55 | 50.67 |
| GPT-4o-mini-0718 | 59.76 | 57.94 | 56.55 | 61.31 | 58.17 | 56.33 | 55.56 | 55.35 | 60.99 | 60.83 |
| GPT-4o-0513 | 70.83 | 61.70 | 59.40 | 61.13 | 62.10 | 68.82 | 56.16 | 56.70 | 57.79 | 59.80 |
| GPT-4o-0806 | 55.14 | 50.31 | 46.00 | 52.15 | 52.45 | 54.13 | 45.43 | 44.25 | 48.26 | 52.18 |
| GPT-4o-1120 | 73.48 | 69.06 | 61.51 | 67.60 | 63.99 | 70.16 | 61.33 | 58.89 | 65.06 | 60.84 |
| Gemini-1.5-Flash | 72.11 | 67.81 | 68.17 | 71.88 | 78.31 | 70.32 | 65.94 | 66.58 | 69.10 | 74.77 |
| Gemini-1.5-Pro | 68.93 | **69.64** | **71.50** | 72.06 | 68.42 | 66.31 | **70.03** | **72.17** | **70.13** | **69.32** |

## C.3 DIFFERENT VERSIONS OF SAME MODEL

We further examine the effect of model capacity on the different metrics of PAIRBENCH. As seen in Figure 11, larger-capacity models tend to perform better across *MMScore*, $\varepsilon$-RelaxSym, and *Cont*. However, there are exceptions—for example, InternVL2-4B demonstrates greater controllability in rotation (R) and perspective shift (PS) compared to InternVL2-8B. Additionally, smoothness (*SM*) does not increase monotonically with model capacity. This suggests that stronger models may be more confident in their responses, leading to less diversity in their similarity scores compared to lower-capacity models.

On the other hand, Table 2 and Figure 13 show that *SM* correlates positively with model performance and other benchmarks, indicating that better models tend to produce smoother and more diverse outputs than weaker ones. Ultimately, we conclude that *SM* is not strictly a property of model performance but rather a characteristic of a VLM as a judge model that may be desirable (or not) depending on the use case.

## C.4 CORRELATIONS

In this section, we further plot the correlations of the different metrics and show them in Figures 12, 13, 13. As seen, all these metrics have positive correlations as seen in the scatter plots.

## C.5 ENCODERS VS VLMS

For the image-image task, we explore how image encoders compare to VLMs on our metrics. To this end, three DINOv2 versions (DINOv2-Base, DINOv2-Small, and DINOv2-Large) and the LAION- and OpenAI- CLIP-trained ViTs (base and large) are chosen to encode images. Since

Table 5: Comparison of the *MMScore* metric ($\times 100$) of VLMs on PB$_{\text{WU-II}}$ (subset A and B) benchmark in the *sensitive* setting. Models are evaluated across multiple criteria: spatial position (SP), spatial position and color jitter (SP-CJ), spatial position and elastic transform (SP-ET), spatial position and gaussian blur (SP-GB), spatial position and perspective shift (SP-PS), and spatial position and rotation (SP-R). Higher scores indicate better performance.

| Model | PAIRBENCH$_{WU_a}$ | | | | | | PAIRBENCH$_{WU_b}$ | | | | | |
|---|---|---|---|---|---|---|---|---|---|---|---|---|
| | SP | SP-CJ | SP-ET | SP-GB | SP-PS | SP-R | SP | SP-CJ | SP-ET | SP-GB | SP-PS | SP-R |
| Chameleon-7B | 00.28 | 00.47 | 00.23 | 00.52 | 0.21 | 00.20 | 00.34 | 00.38 | 00.35 | 00.26 | 00.31 | 00.33 |
| LLaVA-OneVision-7B | 38.95 | 18.83 | 24.03 | 26.78 | 29.46 | 24.63 | 19.70 | 14.03 | 16.51 | 16.78 | 17.76 | 17.02 |
| Phi-3.5-vision | 23.44 | 08.46 | 15.70 | 19.41 | 13.34 | 10.83 | 15.38 | 12.98 | 18.91 | 20.19 | 11.69 | 17.06 |
| Pixtral-12B | 37.91 | 26.09 | 32.05 | 33.52 | 32.47 | 25.00 | 28.02 | 19.58 | 22.32 | 22.31 | 23.46 | 24.50 |
| InternVL2-1B | 00.44 | 00.98 | 00.79 | 00.65 | 00.30 | 00.28 | 00.20 | - | - | 00.41 | 01.18 | 00.90 |
| InternVL2-2B | 22.85 | 12.03 | 14.37 | 17.84 | 18.66 | 15.50 | 20.72 | 10.89 | 11.22 | 15.74 | 17.74 | 13.58 |
| InternVL2-4B | 46.89 | 27.91 | 36.67 | 43.03 | 44.27 | 27.76 | 44.89 | 27.77 | 33.35 | 38.12 | 42.23 | 36.16 |
| InternVL2-8B | 41.99 | 32.06 | 35.71 | 41.02 | 40.12 | 29.11 | 46.36 | 32.17 | 39.24 | 41.90 | 45.59 | 40.30 |
| InternVL2.5-1B | 25.50 | 14.16 | 21.32 | 15.69 | 21.49 | 16.30 | 24.77 | 16.16 | 21.10 | 19.95 | 27.89 | 21.47 |
| InternVL2.5-2B | 20.63 | 11.76 | 16.75 | 15.21 | 18.03 | 13.79 | 23.44 | 09.33 | 15.90 | 17.64 | 18.17 | 17.56 |
| InternVL2.5-4B | 46.15 | 32.74 | 39.05 | 39.24 | 42.28 | 32.94 | 47.93 | 33.75 | 40.23 | 39.82 | 44.07 | 42.57 |
| InternVL2.5-8B | 44.27 | 36.99 | 41.49 | 42.60 | 43.65 | 33.24 | 41.32 | 31.69 | 40.10 | 39.73 | 44.03 | 42.99 |
| MolmoE-1B | 00.47 | 01.03 | 00.00 | 00.03 | 00.14 | 00.01 | 00.32 | 00.36 | 00.01 | 00.04 | 00.04 | 00.09 |
| Molmo-7B-O | 15.94 | 09.90 | 11.32 | 15.38 | 12.92 | 12.01 | 15.15 | 08.40 | 11.39 | 11.33 | 13.60 | 12.50 |
| Molmo-7B-D | 23.82 | 17.75 | 20.41 | 18.40 | 22.21 | 17.81 | 26.74 | 18.37 | 19.55 | 18.77 | 18.19 | 22.21 |
| Qwen2-VL-2B | 02.26 | 01.76 | 02.58 | 02.15 | 03.17 | 01.68 | 00.88 | 00.44 | 00.73 | 00.37 | 00.72 | 00.82 |
| Qwen2-VL-7B | 41.95 | 29.47 | 36.32 | 39.93 | 40.33 | 34.11 | 42.80 | 28.75 | 31.42 | 37.27 | 39.76 | 36.25 |
| GPT-4o-mini-0718 | 42.55 | 37.21 | 39.50 | 40.44 | 38.83 | 41.05 | 48.86 | 38.38 | 43.82 | 45.42 | 46.32 | 46.66 |
| GPT-4o-0513 | 40.27 | 37.83 | 36.79 | 38.52 | 38.84 | 38.07 | 44.13 | 39.46 | 39.46 | 43.58 | 43.49 | 46.25 |
| GPT-4o-0806 | 37.58 | 33.72 | 34.24 | 33.36 | 34.80 | 33.17 | 40.11 | 33.36 | 32.36 | 34.32 | 39.91 | 38.67 |
| GPT-4o-1120 | 40.68 | 39.06 | 40.10 | 40.35 | 40.96 | 40.40 | 47.34 | 40.91 | 43.07 | 47.18 | 50.22 | 50.68 |
| Gemini-1.5-Flash | 44.63 | 38.85 | 37.19 | 39.11 | 35.76 | 34.57 | 49.91 | 40.29 | 42.92 | 46.34 | 47.01 | 46.40 |
| Gemini-1.5-Pro | 40.38 | 36.07 | 31.52 | 37.85 | 29.92 | 30.37 | 49.20 | 38.26 | 39.16 | 44.98 | 41.70 | 40.72 |

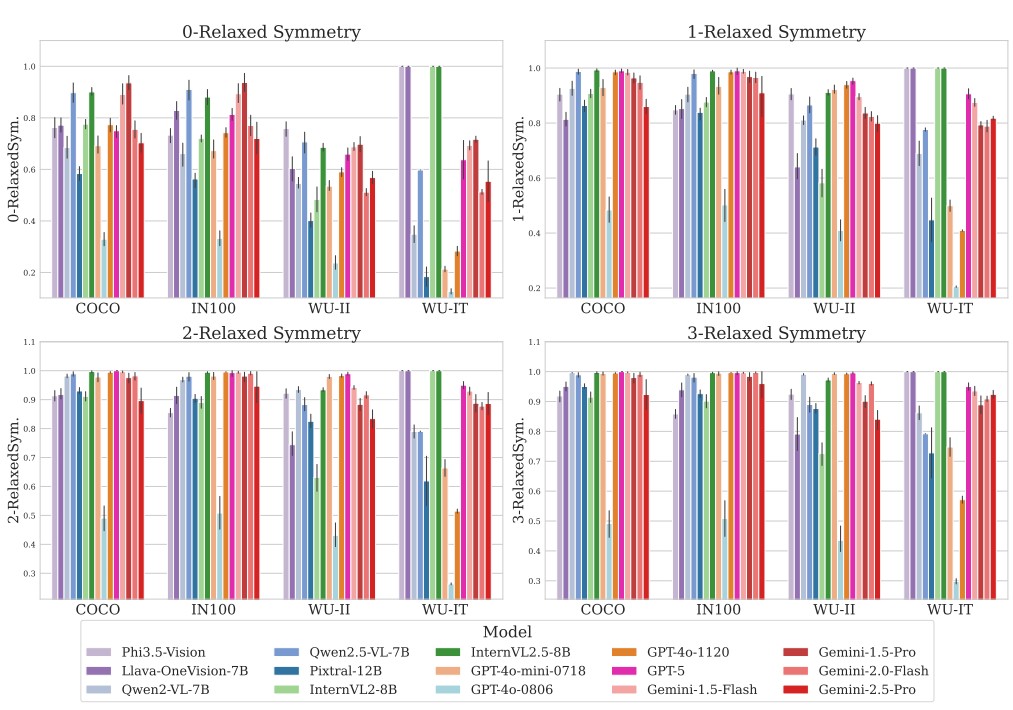

Figure 9: $\varepsilon$-RelaxSym for different $\varepsilon$s.

feature controllability on image-encoders is limited to the image augmentation transformation (CJ, R, PS, GB, ET), we only compare image-encoders to VLMs on PB$_{\text{COCO}}$ and PB$_{\text{IN100}}$.

Table 6: Comparison of the *MMScore* metric ($\times 100$) of VLMs on PB$_{\text{WU-II}}$ (subset A and B) benchmark in the *invariant* setting. Models are evaluated across multiple criteria: spatial position (SP), spatial position and color jitter (SP-CJ), spatial position and elastic transform (SP-ET), spatial position and gaussian blur (SP-GB), spatial position and perspective shift (SP-PS), and spatial position and rotation (SP-R). Higher scores indicate better performance.

| Model | PAIRBENCH$_{WU_a}$ | | | | | | PAIRBENCH$_{WU_b}$ | | | | | |
|---|---|---|---|---|---|---|---|---|---|---|---|---|
| | SP | SP-CJ | SP-ET | SP-GB | SP-PS | SP-R | SP | SP-CJ | SP-ET | SP-GB | SP-PS | SP-R |
| Chameleon-7B | 00.34 | 00.39 | 00.76 | 00.47 | 00.43 | 00.41 | 00.47 | 00.34 | 00.56 | 00.24 | 00.62 | 00.34 |
| LLaVA-OneVision-7B | 34.79 | 31.56 | 30.23 | 34.14 | 32.61 | 28.69 | 13.12 | 18.41 | 16.21 | 22.69 | 15.34 | 17.91 |
| Phi-3.5-vision | 23.66 | 32.84 | 18.90 | 21.36 | 30.14 | 19.10 | 19.88 | 36.74 | 22.40 | 23.47 | 30.04 | 26.06 |
| Pixtral-12B | 36.93 | 37.32 | 41.17 | 35.31 | 38.52 | 36.05 | 36.03 | 30.44 | 33.32 | 29.84 | 35.48 | 33.32 |
| InternVL2-1B | 00.57 | 01.08 | 02.02 | 01.02 | 00.89 | 00.37 | 00.65 | 00.81 | 00.96 | 00.50 | 00.56 | 00.54 |
| InternVL2-2B | 26.25 | 25.53 | 25.76 | 21.12 | 26.57 | 26.98 | 26.03 | 24.52 | 26.49 | 25.81 | 31.01 | 29.33 |
| InternVL2-4B | 39.33 | 40.23 | 37.80 | 42.25 | 43.10 | 34.57 | 51.43 | 41.55 | 45.96 | 50.20 | 54.94 | 50.34 |
| InternVL2-8B | 43.80 | 44.31 | 44.53 | 43.99 | 46.02 | 40.43 | 60.92 | 46.63 | 54.53 | 51.31 | 56.94 | 53.88 |
| InternVL2.5-1B | 12.82 | 13.84 | 09.34 | 07.24 | 12.91 | 16.93 | 19.87 | 24.92 | 19.36 | 17.94 | 22.66 | 30.60 |
| InternVL2.5-2B | 31.38 | 29.79 | 30.53 | 23.16 | 31.75 | 24.69 | 36.01 | 30.13 | 35.52 | 27.07 | 37.01 | 31.18 |
| InternVL2.5-4B | 48.79 | 53.58 | 54.52 | 48.09 | 52.78 | 46.46 | 50.51 | 48.71 | 53.45 | 52.03 | 53.77 | 50.12 |
| InternVL2.5-8B | 59.03 | 55.57 | 59.70 | 57.16 | 58.01 | 50.84 | 65.21 | 51.31 | 61.10 | 63.54 | 62.38 | 60.83 |
| MolmoE-1B | 03.83 | 00.09 | 00.02 | 00.02 | 00.10 | 00.17 | 04.22 | 00.07 | 00.02 | 00.07 | 00.12 | 00.00 |
| Molmo-7B-O | 18.63 | 17.50 | 19.68 | 16.42 | 19.58 | 14.99 | 15.94 | 19.46 | 20.93 | 17.98 | 24.21 | 21.68 |
| Molmo-7B-D | 28.21 | 36.47 | 31.95 | 26.89 | 35.57 | 33.58 | 37.50 | 35.90 | 34.70 | 33.51 | 33.04 | 34.35 |
| Qwen2-VL-2B | 02.63 | 02.88 | 03.58 | 03.53 | 03.34 | 02.97 | 00.79 | 00.73 | 00.99 | 00.88 | 00.71 | 00.82 |
| Qwen2-VL-7B | 40.21 | 38.96 | 39.94 | 46.88 | 40.11 | 39.55 | 47.65 | 39.51 | 40.94 | 48.63 | 44.68 | 41.88 |
| GPT-4o-mini-0718 | 47.60 | 48.33 | 51.04 | 46.15 | 48.86 | 43.75 | 57.50 | 49.19 | 51.38 | 53.76 | 55.82 | 54.07 |
| GPT-4o-0513 | 52.39 | 51.58 | 48.78 | 47.11 | 47.50 | 52.68 | 61.59 | 59.77 | 58.08 | 60.95 | 61.53 | 63.74 |
| GPT-4o-0806 | 50.94 | 47.21 | 46.52 | 42.90 | 45.84 | 52.50 | 62.75 | 54.23 | 53.20 | 51.19 | 58.50 | 57.21 |
| GPT-4o-1120 | 57.47 | 56.25 | 54.40 | 56.11 | 54.40 | 57.93 | 65.91 | 62.22 | 63.93 | 67.96 | 66.86 | 68.10 |
| Gemini-1.5-Flash | 46.62 | 55.28 | 54.31 | 57.98 | 57.01 | 58.74 | 62.04 | 54.43 | 56.89 | 62.24 | 66.88 | 60.72 |
| Gemini-1.5-Pro | 38.07 | 35.08 | 35.05 | 36.11 | 33.21 | 33.23 | 56.43 | 42.24 | 43.74 | 48.41 | 50.40 | 45.83 |

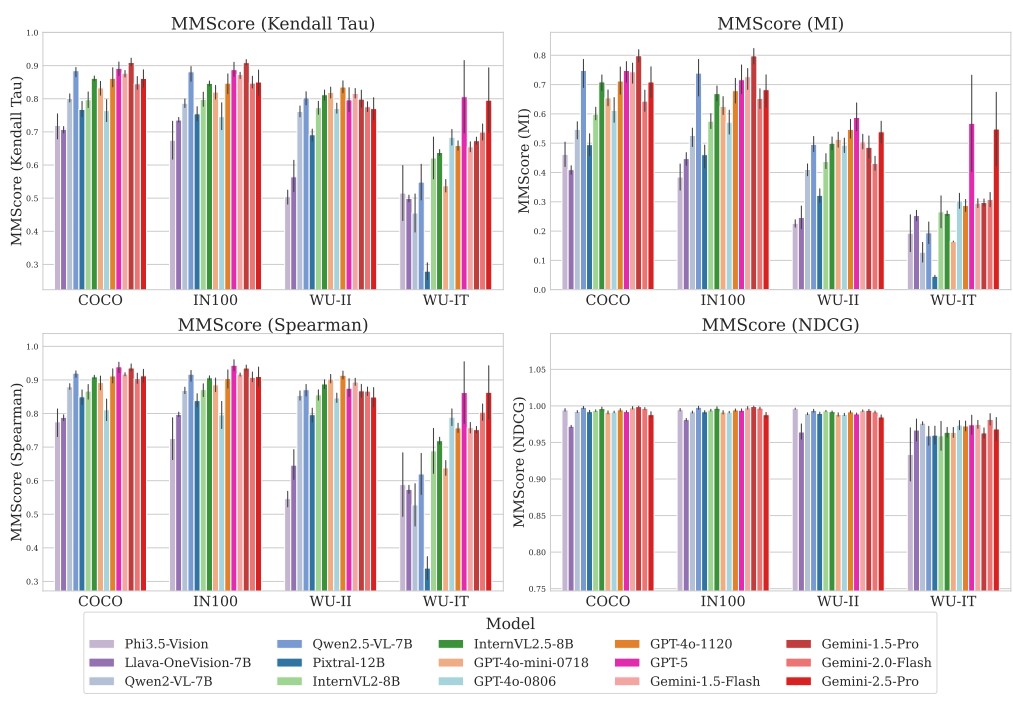

Figure 10: Performance of models on  if computed based on other formulations.

To generate the similarity score of a given image-pair with an image-encoder, we compute the cosine similarity of the representation of each image and scale the scores between 1-10, and round them to the nearest integer. To generate the criteria-sensitive similarity score, we create the representations of the image-pair by simply using the representations output by the encoder for each image. On the

Table 7: Comparison of the *MMScore* metric ($\times 100$) of VLMs on the PB$_{\text{WU-IT}}$ (Subset A and B) benchmark in the *sensitive* and *invariant* settings. Models are evaluated across the spatial position (SP) criterion. Higher scores indicate better performance.

| Model | PAIRBENCH$_{WU_a}$ | | PAIRBENCH$_{WU_b}$ | |
|---|---|---|---|---|
| | **Sens.** | **Invar.** | **Sens.** | **Invar.** |
| Chameleon-7B | 00.25 | 00.34 | 00.23 | 00.47 |
| LLaVA-OneVision-7B | 23.35 | 22.78 | 27.38 | 25.98 |
| Phi-3.5-vision | 13.86 | 12.30 | 25.67 | 24.74 |
| Pixtral-12B | 05.14 | 05.04 | 03.27 | 04.58 |
| InternVL2-1B | 06.29 | 03.75 | 15.90 | 08.31 |
| InternVL2-2B | 17.07 | 14.26 | 24.46 | 16.49 |
| InternVL2-4B | 15.69 | 15.69 | 24.27 | 22.96 |
| InternVL2-8B | 22.40 | 19.27 | 29.45 | 31.46 |
| InternVL2.5-1B | 20.80 | 09.49 | 16.86 | 13.23 |
| InternVL2.5-2B | 15.36 | 11.15 | 19.69 | 18.42 |
| InternVL2.5-4B | 23.90 | 23.85 | 29.75 | 32.45 |
| InternVL2.5-8B | 24.16 | 25.55 | 24.00 | 28.22 |
| MolmoE-1B | 00.12 | 00.04 | 00.02 | 00.21 |
| Molmo-7B-O | 07.53 | 07.45 | 07.18 | 08.29 |
| Molmo-7B-D | 09.45 | 12.26 | 08.34 | 11.26 |
| Qwen2-VL-2B | 02.65 | 03.09 | 05.09 | 05.86 |
| Qwen2-VL-7B | 09.43 | 09.19 | 15.99 | 16.13 |
| GPT-4o-mini-0718 | 16.18 | 16.14 | 16.18 | 15.30 |
| GPT-4o-0513 | 11.49 | 20.48 | 12.63 | 20.98 |
| GPT-4o-0806 | 20.27 | 31.80 | 22.97 | 36.56 |
| GPT-4o-1120 | 18.97 | 31.91 | 20.57 | 34.99 |
| Gemini-1.5-Flash | 27.46 | 26.54 | 26.53 | 32.07 |
| Gemini-1.5-Pro | 26.89 | 27.16 | 28.57 | 29.23 |

other hand, when generating the criteria-invariant score, where the criteria is a specific transformation ($T$), we generate the representation of each image as the average of the representations of the encoder for $k$ versions of the image where random amounts of $T$ are applied to the image. In our experiments, we set $k = 5$.

We report results in Figure 15. We see encoders do better than open-source VLMs most of the time and are comparable to closed-source models (besides CJ). This shows although significantly smaller, encoders can be at least as good as VLMs, enabling similarity scoring at a much lower cost. Also, encoder-generated scores are trivially symmetric as well since the underlying cosine similarity is symmetric. However, they lack in controllability as they are limited to image-only comparisons and can only consider criteria that can be applied to the image using augmentations, i.e., spatial position transform cannot be applied to images for encoders.

## C.6 PROMPT DIVERSITY ANALYSIS

To quantify the impact of prompt phrasing on model performance, we extend the visual analysis presented in Figure 7 (see main text) with the detailed numerical results in Table 8. This table reports the mean MMScore alongside the standard deviation across multiple prompt templates for each evaluated model. The data reveals that sensitivity to prompting is not uniform; the standard deviation varies considerably across different architectures. Furthermore, consistent with the trends observed in Figure 7, no single prompt template yields universally superior performance. These findings highlight

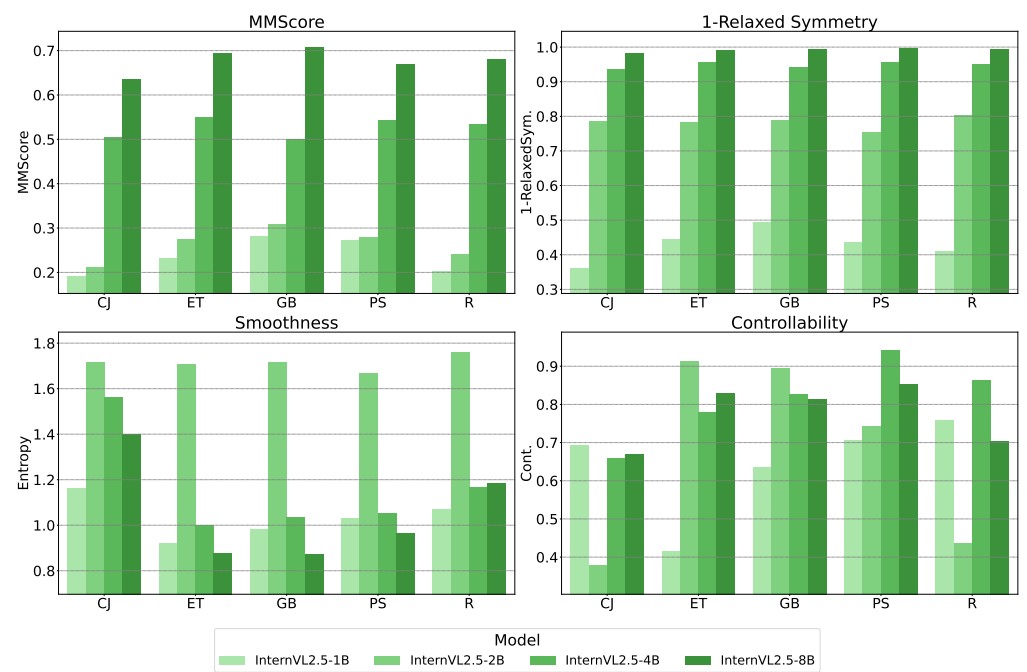

Figure 11: Aggregated PAIRBENCH metrics across different versions of InternVL2.5 models.

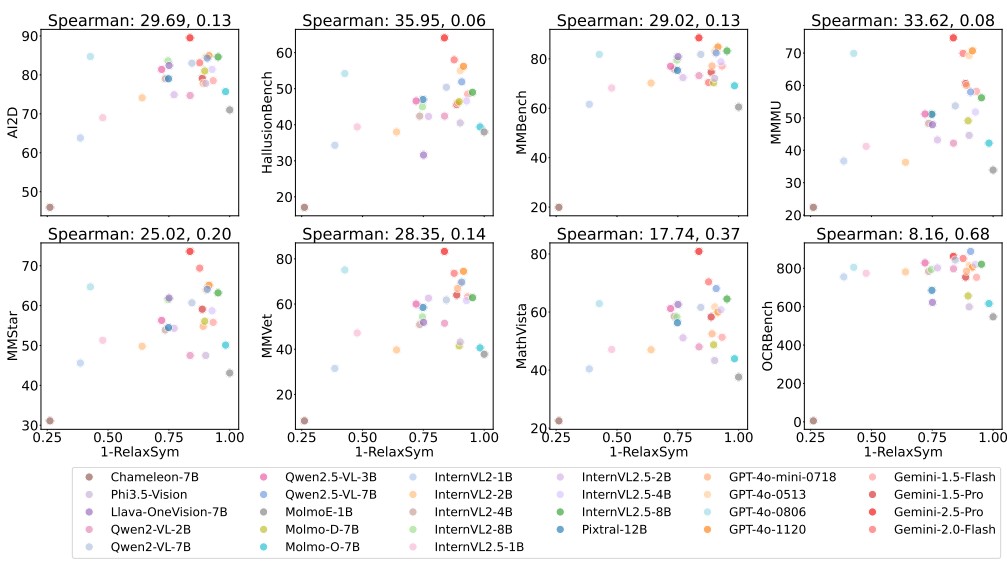

Figure 12: Other benchmarks versus PAIRBENCH on 1-RelaxSym.

the inherent variability in model responses and underscore the necessity of utilizing multiple prompt templates to mitigate selection bias and ensure a robust, fair comparison.

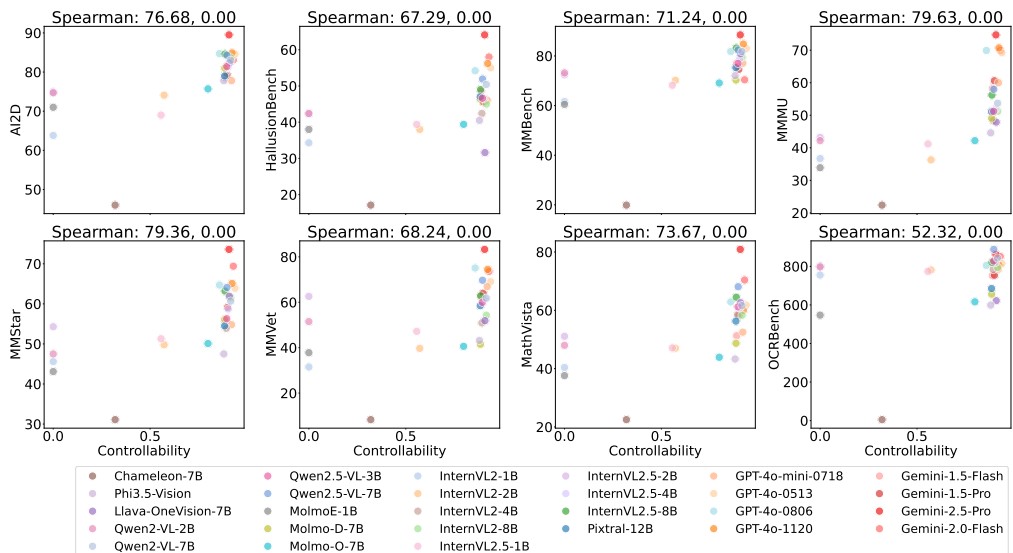

Figure 13: Other benchmarks versus *Cont* on PAIRBENCH.

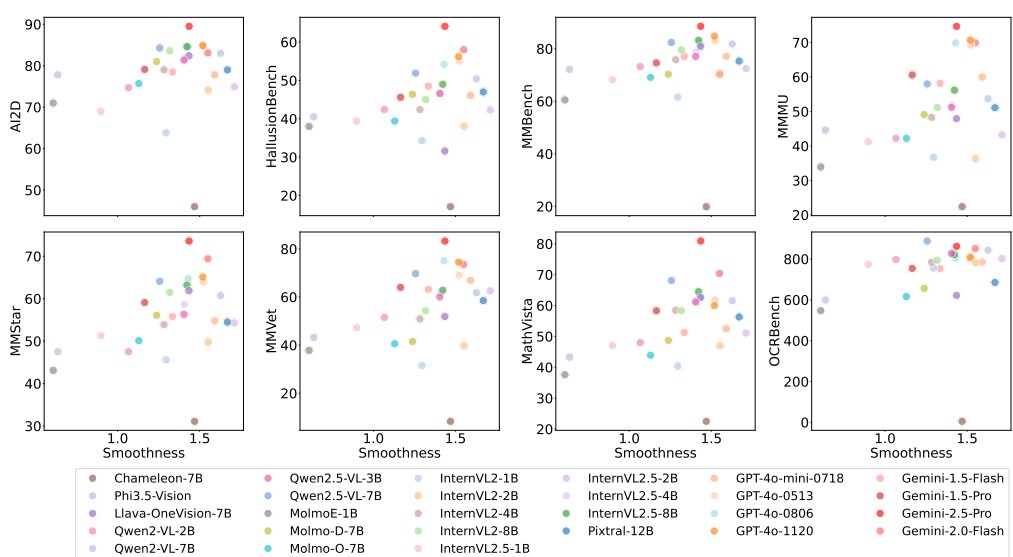

Figure 14: Other benchmarks versus Smoothness (*SM*).

Table 8: Model performance across datasets for different prompt templates. Values are reported as Mean ± Std.

| Model | Dataset | MMScore (Kendall $\tau$) | 0-Relaxed Symmetry | Smoothness | Controllability |
|---|---|---|---|---|---|
| InternVL2-8B | COCO | $80.69\% \pm 1.02\%$ | $76.99\% \pm 5.42\%$ | $1.09 \pm 0.05$ | $89.96\% \pm 2.77\%$ |
| | IN100 | $80.72\% \pm 1.38\%$ | $71.68\% \pm 5.21\%$ | $1.16 \pm 0.03$ | $89.30\% \pm 3.00\%$ |
| | WU-II | $77.98\% \pm 0.64\%$ | $47.27\% \pm 8.46\%$ | $1.47 \pm 0.05$ | $91.75\% \pm 1.26\%$ |
| | WU-IT | $63.11\% \pm 1.06\%$ | $100.00\% \pm 0.00\%$ | $1.05 \pm 0.26$ | $91.65\% \pm 4.29\%$ |

| Model | Dataset | MMScore (Kendall $\tau$) | 0-Relaxed Symmetry | Smoothness | Controllability |
|---|---|---|---|---|---|
| InternVL2.5-8B | COCO | $86.21\% \pm 0.89\%$ | $90.06\% \pm 1.46\%$ | $0.99 \pm 0.03$ | $84.41\% \pm 0.10\%$ |
| | IN100 | $84.71\% \pm 0.64\%$ | $88.07\% \pm 0.68\%$ | $1.11 \pm 0.03$ | $83.30\% \pm 1.25\%$ |
| | WU-II | $81.34\% \pm 0.34\%$ | $68.58\% \pm 1.18\%$ | $1.71 \pm 0.01$ | $90.96\% \pm 1.03\%$ |
| | WU-IT | $64.78\% \pm 2.23\%$ | $100.00\% \pm 0.00\%$ | $1.33 \pm 0.18$ | $96.31\% \pm 1.87\%$ |
| Phi3.5-Vision | COCO | $73.70\% \pm 5.52\%$ | $76.27\% \pm 4.47\%$ | $0.88 \pm 0.16$ | $88.86\% \pm 2.54\%$ |
| | IN100 | $69.08\% \pm 6.68\%$ | $73.31\% \pm 2.95\%$ | $0.80 \pm 0.19$ | $83.72\% \pm 8.25\%$ |
| | WU-II | $52.81\% \pm 9.40\%$ | $75.84\% \pm 4.92\%$ | $0.29 \pm 0.25$ | $74.99\% \pm 7.75\%$ |
| | WU-IT | $52.80\% \pm 2.13\%$ | $100.00\% \pm 0.00\%$ | $0.97 \pm 0.26$ | $95.03\% \pm 2.24\%$ |
| Pixtral-12B | COCO | $78.57\% \pm 3.67\%$ | $58.75\% \pm 9.09\%$ | $1.36 \pm 0.13$ | $87.06\% \pm 2.13\%$ |
| | IN100 | $77.56\% \pm 3.34\%$ | $56.67\% \pm 8.57\%$ | $1.38 \pm 0.11$ | $86.86\% \pm 2.52\%$ |
| | WU-II | $69.53\% \pm 2.16\%$ | $40.26\% \pm 3.63\%$ | $1.81 \pm 0.05$ | $88.64\% \pm 1.37\%$ |
| | WU-IT | $29.35\% \pm 3.46\%$ | $18.29\% \pm 1.90\%$ | $1.93 \pm 0.06$ | $88.31\% \pm 9.29\%$ |
| Qwen2.5-VL-32B | COCO | $83.40\% \pm 1.92\%$ | $78.30\% \pm 2.19\%$ | $1.08 \pm 0.05$ | $87.69\% \pm 1.61\%$ |
| | IN100 | $82.16\% \pm 1.21\%$ | $76.59\% \pm 1.04\%$ | $1.17 \pm 0.06$ | $87.07\% \pm 1.08\%$ |
| | WU-II | $79.59\% \pm 1.48\%$ | $63.66\% \pm 1.96\%$ | $1.76 \pm 0.04$ | $90.78\% \pm 1.41\%$ |
| | WU-IT | $59.21\% \pm 10.09\%$ | $26.81\% \pm 24.50\%$ | $1.75 \pm 0.12$ | $89.51\% \pm 13.00\%$ |
| Qwen2.5-VL-3B | COCO | $73.55\% \pm 1.37\%$ | $73.94\% \pm 1.03\%$ | $1.40 \pm 0.06$ | $84.87\% \pm 3.40\%$ |
| | IN100 | $75.07\% \pm 2.79\%$ | $74.06\% \pm 1.93\%$ | $1.31 \pm 0.07$ | $81.89\% \pm 2.19\%$ |
| | WU-II | $57.15\% \pm 1.38\%$ | $42.48\% \pm 2.51\%$ | $1.44 \pm 0.03$ | $91.71\% \pm 1.82\%$ |
| | WU-IT | $37.94\% \pm 1.78\%$ | $51.49\% \pm 8.77\%$ | $1.02 \pm 0.12$ | $87.69\% \pm 11.31\%$ |
| Qwen2.5-VL-7B | COCO | $88.56\% \pm 0.31\%$ | $89.80\% \pm 1.57\%$ | $0.99 \pm 0.04$ | $87.52\% \pm 1.10\%$ |
| | IN100 | $88.22\% \pm 0.36\%$ | $91.06\% \pm 0.96\%$ | $0.96 \pm 0.04$ | $85.68\% \pm 1.48\%$ |
| | WU-II | $80.42\% \pm 0.54\%$ | $70.66\% \pm 1.77\%$ | $1.46 \pm 0.03$ | $91.01\% \pm 1.33\%$ |
| | WU-IT | $55.61\% \pm 2.52\%$ | $59.81\% \pm 3.50\%$ | $1.24 \pm 0.10$ | $90.53\% \pm 5.49\%$ |
| Qwen2-VL-7B | COCO | $80.83\% \pm 1.79\%$ | $68.48\% \pm 2.26\%$ | $1.43 \pm 0.05$ | $91.14\% \pm 1.43\%$ |
| | IN100 | $80.23\% \pm 2.15\%$ | $66.16\% \pm 2.79\%$ | $1.38 \pm 0.03$ | $92.24\% \pm 2.34\%$ |
| | WU-II | $76.77\% \pm 0.45\%$ | $54.57\% \pm 2.18\%$ | $1.75 \pm 0.06$ | $89.76\% \pm 1.34\%$ |
| | WU-IT | $47.45\% \pm 2.84\%$ | $34.87\% \pm 3.40\%$ | $1.55 \pm 0.10$ | $94.09\% \pm 3.01\%$ |
| GPT-4o-mini-0718 | COCO | $84.37\% \pm 1.14\%$ | $69.27\% \pm 2.91\%$ | $1.16 \pm 0.04$ | $90.04\% \pm 0.80\%$ |
| | IN100 | $83.15\% \pm 0.86\%$ | $67.28\% \pm 2.67\%$ | $1.20 \pm 0.03$ | $89.14\% \pm 1.43\%$ |
| | WU-II | $82.45\% \pm 0.39\%$ | $53.52\% \pm 2.56\%$ | $1.81 \pm 0.03$ | $92.53\% \pm 1.10\%$ |
| | WU-IT | $53.77\% \pm 6.44\%$ | $21.24\% \pm 5.63\%$ | $2.11 \pm 0.05$ | $95.72\% \pm 2.78\%$ |
| GPT-4o-0513 | COCO | $85.54\% \pm 1.09\%$ | $73.40\% \pm 1.51\%$ | $1.03 \pm 0.04$ | $93.29\% \pm 0.48\%$ |
| | IN100 | $83.80\% \pm 1.20\%$ | $70.65\% \pm 1.64\%$ | $1.12 \pm 0.02$ | $92.92\% \pm 0.92\%$ |
| | WU-II | $82.75\% \pm 0.28\%$ | $53.42\% \pm 0.81\%$ | $1.78 \pm 0.01$ | $94.73\% \pm 0.49\%$ |
| | WU-IT | $52.47\% \pm 4.86\%$ | $22.39\% \pm 3.87\%$ | $1.99 \pm 0.08$ | $88.03\% \pm 7.59\%$ |
| GPT-4o-0806 | COCO | $73.70\% \pm 4.35\%$ | $24.38\% \pm 18.29\%$ | $0.95 \pm 0.05$ | $82.50\% \pm 5.80\%$ |
| | IN100 | $72.66\% \pm 3.23\%$ | $24.73\% \pm 18.83\%$ | $0.98 \pm 0.07$ | $79.32\% \pm 7.92\%$ |
| | WU-II | $74.34\% \pm 5.40\%$ | $18.06\% \pm 12.27\%$ | $1.62 \pm 0.08$ | $85.94\% \pm 4.17\%$ |
| | WU-IT | $69.06\% \pm 0.91\%$ | $11.89\% \pm 9.10\%$ | $2.00 \pm 0.09$ | $85.32\% \pm 7.09\%$ |
| GPT-4o-1120 | COCO | $86.58\% \pm 1.00\%$ | $77.32\% \pm 1.93\%$ | $1.04 \pm 0.03$ | $91.89\% \pm 1.23\%$ |
| | IN100 | $85.10\% \pm 1.35\%$ | $74.24\% \pm 2.39\%$ | $1.11 \pm 0.04$ | $91.45\% \pm 0.37\%$ |
| | WU-II | $83.82\% \pm 0.33\%$ | $59.05\% \pm 1.08\%$ | $1.77 \pm 0.03$ | $93.85\% \pm 0.61\%$ |
| | WU-IT | $65.78\% \pm 2.80\%$ | $28.51\% \pm 4.42\%$ | $1.92 \pm 0.09$ | $79.85\% \pm 5.87\%$ |

Table 8 – continued from previous page

| Model | Dataset | MMScore (Kendall $\tau$) | 0-Relaxed Symmetry | Smoothness | Controllability |
|---|---|---|---|---|---|
| | | **Table 8 – continued from previous page** | | | |
| Gemini-1.5-Flash | COCO | $87.75\% \pm 0.63\%$ | $89.08\% \pm 0.71\%$ | $0.87 \pm 0.02$ | $85.41\% \pm 1.08\%$ |
| | IN100 | $87.40\% \pm 0.27\%$ | $89.38\% \pm 1.06\%$ | $0.91 \pm 0.05$ | $83.86\% \pm 1.01\%$ |
| | WU-II | $81.72\% \pm 0.31\%$ | $68.75\% \pm 1.43\%$ | $1.60 \pm 0.03$ | $91.44\% \pm 1.09\%$ |
| | WU-IT | $66.40\% \pm 4.17\%$ | $69.39\% \pm 3.51\%$ | $1.70 \pm 0.13$ | $94.32\% \pm 2.56\%$ |
| Gemini-2.0-Flash | COCO | $85.79\% \pm 2.56\%$ | $75.57\% \pm 8.20\%$ | $1.09 \pm 0.13$ | $90.45\% \pm 1.36\%$ |
| | IN100 | $85.91\% \pm 2.37\%$ | $76.99\% \pm 6.52\%$ | $1.10 \pm 0.09$ | $89.44\% \pm 1.06\%$ |
| | WU-II | $78.08\% \pm 0.96\%$ | $51.18\% \pm 1.14\%$ | $1.81 \pm 0.03$ | $93.78\% \pm 1.02\%$ |
| | WU-IT | $71.12\% \pm 2.08\%$ | $51.23\% \pm 4.22\%$ | $1.71 \pm 0.05$ | $95.29\% \pm 2.50\%$ |
| Gemini-1.5-Pro | COCO | $90.97\% \pm 0.50\%$ | $93.58\% \pm 1.68\%$ | $0.76 \pm 0.04$ | $83.31\% \pm 1.47\%$ |
| | IN100 | $90.99\% \pm 0.79\%$ | $93.72\% \pm 0.75\%$ | $0.76 \pm 0.02$ | $83.49\% \pm 1.42\%$ |
| | WU-II | $80.40\% \pm 1.97\%$ | $69.79\% \pm 2.64\%$ | $1.41 \pm 0.08$ | $93.26\% \pm 3.55\%$ |
| | WU-IT | $68.74\% \pm 3.71\%$ | $71.81\% \pm 6.28\%$ | $1.37 \pm 0.12$ | $93.86\% \pm 2.05\%$ |

## D    PAIRBENCH DETAILS

### D.1    DATASET CREATION

The PAIRBENCH framework takes in a source dataset and creates augmented versions of the data to obtain data pairs to probe the evaluation skills of a model. In our instances, we use COCO (Lin et al., 2014), IN100 (Deng et al., 2009) and WhatsUp (Kamath et al., 2023) datasets as the source for the original data points. We utilize COCO and IN100 as image-only datasets and WhatsUp as an image-text dataset. We select 500 random images from each of COCO and IN100 and all the image-text pairs from both subsets provided by the WhatsUp dataset to be used in our instantiation of PAIRBENCH. Full details of our released datasets are given in Table 9.

To isolate the effect of different data characteristics on model performance, PAIRBENCH creates pairs of image-image and image-text data that are identical except for one or a few controlled features. The generated data consists of points from the original dataset paired with their transformed version. For COCO and IN100, we create a different control sample for each one of the transformations in {color jitter, rotation, gaussian blur, perspective shift, elastic transformation}, which defines the characteristic that differs between images. For the data from WhatsUp, we construct the data pairs by either only using the 'spatial position' transform, or 'spatial position' transform in addition to one of the previous five characteristics to additionally assess coupling effects. However, since transforms are not well-defined for texts, only 'spatial position' transform is applied for the image-text pairs. Note that the image-image pairs from WhatsUp are the most challenging since they all have at least the 'spatial position' transform, which is a well-known blind-spot of VLMs as shown by previous literature (Kamath et al., 2023; Wang et al., 2024a). As a result, we end up creating five image-image sub-datasets for each of COCO and IN100, six subsets for each of the two subsets of WhatsUp, using each of the transformations, and one image-text sub-dataset for each of the subsets of WhatsUp. The details of the transforms applied to each category are shown in Figure 3.

Next, for each original image, we construct three types of pairs: an identical, a transformed, and an irrelevant pair. In all three versions of these pairs, the first data point is the original (non-transformed) image. For the 'identical' pair, the second data point is another version of the image with 95% of its original size for the image-image pair and the correct caption for the image-text pair. The second data point in the 'transformed' pair is the original image (caption) with the transformation applied to it for the image-image (image-text) pair. Finally, the 'irrelevant' pair's second data point is a transformed version of a random image (caption) from the rest of the dataset.

Equipped with the constructed control samples, PAIRBENCH prompts the VLM to score the similarity of each data pair based on a set of criteria. The criteria consists of the conditions indicating whether the model under examination should be 'sensitive' or 'invariant' to the transformations applied for that specific sub-dataset. These two settings (sensitive or invariant) measure how well each model

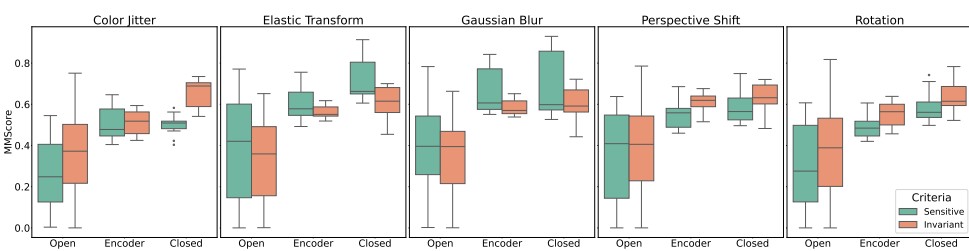

Figure 15: A simple vision encoder outperforms open-sourced VLMs and has on par performance with closed sourced models which are much more expensive, for image-image tasks (results combine PB$_{\text{COCO}}$ and PB$_{\text{IN100}}$), and similar pattern is observed across different transformations.

Table 9: Information of different splits in PB$_{\text{COCO}}$, PB$_{\text{IN100}}$, PB$_{\text{WU-II}}$, and PB$_{\text{WU-IT}}$.

| Modality | Source | Number of Selected | Splits | New Data Points / Total Data-Pair Comparisons |
|---|---|---|---|---|
| PB$_{\text{II}}$ | COCO | 500 | CJ | 1000 / 3000 |
| | | | R | 1000 / 3000 |
| | | | ET | 1000 / 3000 |
| | | | PS | 1000 / 3000 |
| | | | GB | 1000 / 3000 |
| | IN100 | 500 | CJ | 1000 / 3000 |
| | | | R | 1000 / 3000 |
| | | | ET | 1000 / 3000 |
| | | | PS | 1000 / 3000 |
| | | | GB | 1000 / 3000 |
| | WhatsUp (subset A) | 418 | SP | 0 / 3344 |
| | | | SP & CJ | 1254 / 3344 |
| | | | SP & R | 1254 / 3344 |
| | | | SP & ET | 1254 / 3344 |
| | | | SP & PS | 1254 / 3344 |
| | | | SP & GB | 1254 / 3344 |
| | WhatsUp (subset B) | 408 | SP | 0 / 3264 |
| | | | SP & CJ | 1224 / 3264 |
| | | | SP & R | 1224 / 3264 |
| | | | SP & ET | 1224 / 3264 |
| | | | SP & PS | 1224 / 3264 |
| | | | SP & GB | 1224 / 3264 |
| PB$_{\text{IT}}$ | WhatsUp (subset A) | 418 | SP | 1254 / 3344 |
| | WhatsUp (Subset B) | 408 | SP | 1224 / 3264 |
| **In total** | - | 1826 | all splits | **22390 / 69648** |

can recognize the differences between the data pair and follow the prompt's criteria. If a model can successfully capture a specific feature, it will have no problem being variant or invariant to it; however, if it cannot detect it or has a bias towards a feature, it will favor being sensitive or invariant to that feature over its opposite. Using a human study, described in Appendix D.2, the ground-truth score of the 'identical' and 'irrelevant' pair are set to 10 and 1, respectively, in both 'sensitive' and 'invariant' settings. However, for the 'transformed' pair, based on the human study we set the score 10 in the 'invariant' version, and '6' in the 'sensitive' version of the prompt. To make sure the performance gap between models is not merely a consequence of biased prompt wording, PAIRBENCH comes with five template prompts with different lengths and wordings but with the same semantic meaning, that are randomly selected for each data pair, to make sure the prompting does not affect the model's performance. These prompt templates are reported in Appendix D.3.

Ultimately, we end up with 4 different datasets created by PAIRBENCH: PB$_{\text{COCO}}$, PB$_{\text{IN100}}$, PB$_{\text{WU-II}}$, and PB$_{\text{WU-IT}}$. PB$_{\text{COCO}}$ and PB$_{\text{IN100}}$ compare and score image-pairs and have 5 splits (Color Jitter

(CJ), Rotation (R), Gaussian Blur (GB), Perspective Shift (PS), and Elastic Transformation (ET). $PB_{\text{WU-II}}$ consists of 2 subsets, each with 6 splits; one split with only the Spatial Position transform (SP), and the rest with SP combined with one of the previous five transformations (CJ, R, GB, PS, and ET). $PB_{\text{WU-IT}}$ consists of only the SP split for each of the two subsets in the WhatsUp dataset. Details of each split in Appendix D.

## D.2 HUMAN STUDY FOR GROUND TRUTH SCORES

To validate the alignment between our ground-truth scores and human perception, we conducted a human study on **image-image** pairs from $PB_{\text{IN100}}$. We excluded image-text comparisons due to their trivial nature for human judgment. For example, given an image showing a book to the left of a cap, comparing it to the sentence "book left of cap" (identical), "book right of cap" (transformed), or "can behind candle" (irrelevant) would result in nearly unanimous responses, offering limited insight.

Our study involved 76 volunteer participants and covered 300 image pairs sampled across three transformation types—color jitter, perspective shift, and rotation, under both "sensitive" and "invariant" settings. Results from this evaluation led us to adjust the ground-truth score of transformed pairs in the "sensitive" condition to 6 (on a 1–10 scale), as this better captured the perceptual similarity reported by humans. Furthermore, the study confirmed that "identical" pairs consistently received the highest scores, while "irrelevant" pairs received the lowest, supporting the validity of our scoring protocol. A screenshot of the study can be seen in Figure 16 and the results are reported in Table 10.

To assess the consistency of these judgments and account for potential subjectivity, we calculated Krippendorff's alpha Hayes & Krippendorff (2007) on the gathered human data following recent methodologies Wiles et al. (2024); Hu et al. (2023). We obtained an alpha of 0.9396, which indicates strong reliability in the conclusions drawn from the study. Additionally, we measured the leave-one-rater-out noise ceiling using Spearman correlation and achieved an average of 0.9166. This further confirms that the raters exhibit high agreement and that individual human judgments are highly predictable from the average of the group.

Table 10: Human similarity scores (mean ± std) across different transformation settings, which we used to set our ground truth scores.

| Pair Type | Colorjitter | | Perspective | | Rotate | |
|---|---|---|---|---|---|---|
| | **Sens** | **Invar** | **Sens** | **Invar** | **Sens** | **Invar** |
| Identical | 9.8 ± 0.53 | 9.95 ± 0.23 | 9.82 ± 0.42 | 9.9 ± 0.32 | 10.0 ± 0.0 | 9.89 ± 0.51 |
| Transformed | 5.5 ± 2.3 | 9.68 ± 0.85 | 6.89 ± 1.97 | 9.06 ± 1.33 | 6.31 ± 2.01 | 9.36 ± 1.01 |
| Irrelevant | 1.35 ± 0.74 | 1.42 ± 0.67 | 1.33 ± 0.81 | 1.61 ± 1.17 | 1.27 ± 0.64 | 1.23 ± 0.47 |

# PairBench Human Evaluation

👤 **User ID:** `testing`

> **Do not refresh page while taking the survey!**

## Sample 1 of 30

**Instruction:**

Score the similarity of the two images **on a scale of 1 (least similar) to 10 (completely similar)** given the condition[s] below:

- **- Be invariant to rotations while evaluating images. For example, if one image is a rotated version of the other, the similarity score should remain high. If the images are different, the similarity score should decrease regardless of their orientation.**

*Note: Identical images that do not meet the condtion[s] should score **higher** than irrelevant images.*

⚡ *Similarity conditions for **this** pair:*

- *Rotational difference **DOES NOT decrease** the score.*

*If the images are different, decrease the score regardless.*

Image 1     Image 2

**Select your score (1 = least similarity, 10 = highest similarity):**

| 1 | 2 | 3 | 4 | 5 | 6 | 7 | 8 | 9 | 10 |

🔁 Restart with new samples

Progress: 1 / 30

Figure 16: Example screenshot of the platform used consisting of the type of questions that participants were asked during the study. In the current sample, although both images depict birds, given that they are not a transformation (e.g., rotation) of each other, they represent an irrelevant pair; so the ground truth score would be the minimum score.

### D.3 PROMPT TEMPLATES FOR DIFFERENT PAIRBENCH DATASETS

We provide the 5 different templates that we choose at random for each data pair for the image-image and image-text prompts. The following are the templates we utilize for $PB_{COCO}$ and $PB_{IN100}$, and $PB_{WU-II}$, i.e., the image-image pairs.

---

**Image-Image Prompt Template V1**

*User prompt*: You are tasked with evaluating the similarity between two images while paying attention to the following conditions: {conditions}. Your goal is to judge the similarity of the images overall, where satisfying the conditions increases the similarity score. If the images are identical but fail to meet any of the conditions, they should still receive a higher score than completely unrelated images. Provide a similarity score on a scale from 1 to 10, where 1 represents entirely dissimilar images and 10 represents identical images that satisfy all conditions. Ensure your response is strictly in the following format:

```
Score: <1-10>
Reason: <reason for score>
```

Do not include anything else in your response. What score would you assign to this pair of images? "data1" "data2".

---

**Image-Image Prompt Template V2**

*User prompt*: As a similarity evaluator, your responsibility is to assess the similarity of the given images while considering these conditions: {conditions}. The similarity score should reflect both how well the images align with the conditions and their overall resemblance. Images that are identical but do not meet the conditions should receive a moderate score, while completely unrelated images should receive the lowest score. Provide your score on a scale of 1 to 10, with 10 being identical images that fully meet the conditions. Ensure your response is in the following format:

```
Score: <1-10>
Reason: <reason for score>
```

Provide nothing else. What is your score? "data1" "data2"

---

**Image-Image Prompt Template V3**

*User prompt*: Evaluate the similarity of the images based on the following conditions: {conditions}. The score should take into account how well the images align with these conditions, as well as their overall resemblance. Even if the images are identical but fail to meet the conditions, they should still receive a higher score than completely different images. Provide a score from 1 to 10, where 1 indicates no similarity and 10 indicates identical images that fully satisfy the conditions. Respond only in this format:

```
Score: <1-10>
Reason: <reason for score>
```

Nothing else should be included. What score would you give? "data1" "data2"

---

---

**Image-Image Prompt Template V4**

*User prompt*: Judge the similarity of these images based on: {conditions}. The similarity score should reflect both the overall resemblance of the images and how well they satisfy the conditions. Identical images that do not meet the conditions should still score higher than completely unrelated images. Provide a score on a scale of 1 to 10, with 1 being no similarity and 10 being identical images that satisfy all conditions. Respond strictly in this format:

```
        Score: <1-10>
        Reason: <reason for score>
```

Do not include additional text. What's your rating? "data1" "data2"

---

**Image-Image Prompt Template V5**

*User prompt*: Rate the similarity of these images on a scale of 1 to 10 while considering {conditions}. The score should account for both how well the images satisfy the conditions and their overall resemblance. If the images are identical but do not meet the conditions, they should still score higher than unrelated images. Your response must follow this exact format:

```
        Score: <1-10>
        Reason: <reason for score>
```

Provide nothing else. What's your score? "data1" "data2"

---

And these are the image-text versions for PB$_{\text{WU-IT}}$:

---

**Image-Text Prompt Template V1**

*User prompt*: You are assigned the task of determining how accurately a text description represents an image of two objects. Pay special attention to the following conditions: {conditions}. Your goal is to assess the alignment between the image and the text, where adherence to the conditions increases the similarity score. If the text partially describes the image but fails to meet specific conditions, it should still score higher than a text entirely unrelated to the image. Assign a similarity score on a scale from 1 to 10, where 1 signifies no correlation, and 10 denotes perfect alignment satisfying all conditions. The data pair (text and image) will follow this prompt. Respond strictly in the following format:

```
        Score: <1-10>
        Reason: <reason for score>
```

What score would you assign to this text-image pair? "data1" "data2".

---

**Image-Text Prompt Template V2**

*User prompt*: Figure out how well this image matches the description provided. The image shows two objects, and the text is meant to describe how they're arranged. Look at these specific conditions: {conditions}. If the text captures some parts of the image but misses others, it should still get a better score than something totally off. Score this match on a scale of 1 to 10, where 1 means there's no match and 10 means the description nails it and matches every condition perfectly. The text and image will follow this prompt. Answer in this format only:

```
        Score: <1-10>
        Reason: <reason for score>
```

What's your score? "data1" "data2".

---

---

**Image-Text Prompt Template V3**

*User prompt*: Evaluate the degree to which a text description accurately represents an image featuring two objects, taking into account the following conditions: `{conditions}`. Assign a score based on how well the image-text pair matches, where: - A perfect description that satisfies all conditions scores 10. - Texts that partially align with the image but fail to meet conditions should still score higher than completely unrelated ones. The data pair will follow this prompt. Provide your score on a scale of 1 to 10 using the exact format below:

```
Score: <1-10>
Reason: <reason for score>
```

What score would you give? `"data1"` `"data2"`.

---

**Image-Text Prompt Template V4**

*User prompt*: You are tasked with reviewing how well a text description aligns with an image of two objects. The score should reflect not only the accuracy of the alignment but also how well the description satisfies the following conditions: `{conditions}`. Even if the text description captures some parts of the image while failing the conditions, it should still receive a higher score than a completely irrelevant description. The text and image will be provided below. Assign a score on a 1 to 10 scale, where 1 is no similarity and 10 is perfect alignment that meets all conditions. Answer only in this format:

```
Score: <1-10>
Reason: <reason for score>
```

What score would you assign? `"data1"` `"data2"`.

---

**Image-Text Prompt Template V5**

*User prompt*: Assess the degree to which a text description corresponds to an image of two objects, taking into account the following conditions: `{conditions}`. The scoring should reflect: - A perfect alignment with the image that satisfies all conditions merits a score of 10. - Descriptions that partially match the image but fail to meet certain conditions should still receive a higher score than entirely unrelated descriptions. - A score of 1 should be reserved for cases where no correlation exists between the text and the image. The text and image pair will be provided below. Provide your evaluation using the following format:

```
Score: <1-10>
Reason: <reason for score>
```

What score would you assign? `"data1"` `"data2"`.

## E  LLM USAGE

LLMs were used in this work as assistive tools, but did not contribute as co-authors. Their usage was limited to the following areas:

1. **Benchmark Evaluations:** Since PairBench is a benchmark paper, we employed LLMs as evaluators to score and rank model outputs under different conditions.

2. **Automation of LaTeX Tables and Editing:** LLMs were used to automate the generation of LaTeX tables summarizing results and ablations, as well as for minor editing tasks (e.g., reformatting sections, ensuring consistent style).

3. **Writing Assistance:** LLMs assisted with grammar checking, improving sentence clarity, and smoothing transitions between sections. All scientific claims, analyses, and conclusions were written and verified by the human authors.

