# OpenReview forum: "PairBench: Are Vision-Language Models Reliable at Comparing What They See?"
_ICLR.cc/2026/Conference — Submitted to ICLR 2026_

### Official Review · Reviewer_Jopj · 2025-10-26

**Soundness:** 2
**Presentation:** 3
**Contribution:** 2
**Rating:** 4
**Confidence:** 4

**Summary:**

This paper introduces **PAIRBENCH**, a systematic framework for evaluating the comparison abilities of vision-language models (VLMs), together with four key metrics, including alignment with human, order-invariance, score diversity and instruction following. The benchmark uses existing datasets (ImageNet, COCO, and WhatsUp) to create identical, transformed, and irrelevant image pairs, with human annotations defining ground-truth similarity. The authors benchmark a wide range of open and closed models (GPT, Gemini, InternVL, Qwen, LLaVA, etc.) and show that no model performs consistently across all metrics.

**Strengths:**

* PAIRBENCH fills a clear gap in multimodal evaluation by focusing on pairwise relational reliability rather than standard task-specific benchmarks.
* The use of both invariant and sensitive prompt conditions provides a nuanced test of controllability and instruction following.
* Correlation analysis with external benchmarks (AI2D, HallusionBench, MMMU, etc.) strengthens the empirical insight that comparison ability predicts higher-level reasoning.

**Weaknesses:**

* The framework assumes that human similarity judgments are the gold standard but does not discuss possible variability or subjectivity in those scores. Human judgments can vary substantially across individuals, and there is a lack of such analysis.
* The benchmark’s image pairs are generated through basic transformations (rotation, crop, color jitter). These transformations test low-level invariance rather than semantic equivalence. Real-world comparisons often involve semantically similar but visually diverse images, such as cross-domain or stylistically different examples. As a result, PAIRBENCH may not accurately reflect the reliability of VLMs in practical use cases (e.g., retrieval or multimodal evaluation).
* The paper does not explore using image generation models (e.g., Stable Diffusion, DALL·E) to create semantically similar but distributionally distinct image pairs. Such generative approaches could produce more realistic and diverse comparisons. This would help test whether VLMs capture conceptual similarity rather than superficial pixel-level correspondence
* No systematic study of which types of transformations or linguistic prompts most affect model performance.

**Questions:**

You mention that prompt phrasing significantly affects MMScore. Could you provide quantitative results on how large this variance is (e.g., standard deviation across prompts)?

---

> ### Author Response · Authors · 2025-11-20
> **Response 1/2**
>
> Thank you for the review. I address all noted weaknesses in separate items below.
>
> - "*The framework assumes that human similarity judgments are the gold standard but does not discuss possible variability or subjectivity in those scores. Human judgments can vary substantially across individuals, and there is a lack of such analysis.*"
>   - **Response:** Thank you for your comment. To address the reviewer's concern, we calculated the Krippendorff's alpha [1] following recent work [2,3] for the gathered human data, and it was 0.9396 (the closer to 1, the better), which shows that the conclusions drawn from the human study are reliable. We also measured the leave-one-rater-out noise ceiling using the Spearman correlation and achieved an average of 0.9166, once again confirming that the raters agree with each other and that the left-out human is indeed highly predictable from the average of the rest of the group. We appended the explanations regarding the inter-rater reliability and leave-one-out experiment to Appendix D.2.
>
> [1] Hayes AF, Krippendorff K. Answering the call for a standard reliability measure for coding data. Communication methods and measures. 2007 Apr 1;1(1):77-89.
>
> [2] Wiles O, Zhang C, Albuquerque I, Kajić I, Wang S, Bugliarello E, Onoe Y, Papalampidi P, Ktena I, Knutsen C, Rashtchian C. Revisiting text-to-image evaluation with gecko: On metrics, prompts, and human ratings. arXiv preprint arXiv:2404.16820. 2024 Apr 25.
>
> [3] Hu Y, Liu B, Kasai J, Wang Y, Ostendorf M, Krishna R, Smith NA. Tifa: Accurate and interpretable text-to-image faithfulness evaluation with question answering. In Proceedings of the IEEE/CVF International Conference on Computer Vision 2023 (pp. 20406-20417).
>
> ---
> - "*The benchmark's image pairs are generated through basic transformations (rotation, crop, color jitter). These transformations test low-level invariance rather than semantic equivalence. Real-world comparisons often involve semantically similar but visually diverse images, such as cross-domain or stylistically different examples. As a result, PAIRBENCH may not accurately reflect the reliability of VLMs in practical use cases (e.g., retrieval or multimodal evaluation).*"
>   - **Response:** Thank you for the comment. We agree that the transformations in PairBench do not cover the full space of real-world semantic variation, and this is intentional. PairBench is not designed as a semantic similarity or object-identity benchmark, but as a controlled diagnostic tool for evaluating whether a model can detect specific visual changes, follow instructed invariances or sensitivities, and assign consistent similarity scores. These simple and well-defined transformations provide clear ground truth, which is not feasible when comparing semantically similar but visually diverse images. Our focus is on assessing reliability for automatic judging and reward-model settings, where consistency, symmetry, smoothness, and controllability are critical and not captured by existing multimodal benchmarks. PairBench is therefore complementary to semantic similarity evaluations, and we will clarify this motivation and note that more semantically rich comparison tasks are an interesting direction for future work.
> ---
> - "*The paper does not explore using image generation models (e.g., Stable Diffusion, DALL·E) to create semantically similar but distributionally distinct image pairs. Such generative approaches could produce more realistic and diverse comparisons. This would help test whether VLMs capture conceptual similarity rather than superficial pixel-level correspondence*"
>   - **Response:** Thank you for the suggestion. Please note that we intentionally use simple and fully controlled transformations. This is because those allow unambiguous ground truth and let us isolate basic but essential comparison skills, which is the goal of our benchmark. The whole point of our paper is to show that simple controlled transformations provide strong signals about model behavior and expose limitations in how VLMs perform visual comparisons.
>
>     We agree that once models reliably handle these foundational tasks, expanding PairBench with semantic, distributionally diverse, or generatively produced pairs will be a natural next step for assessing higher-level similarity, temporal variation, and complex multi-object scenarios. But as it is, models already struggle, and PairBench can inform next iterations of models so we can get VLMs able to compare more reliably, which is a fundamental skill we need in several applications.

---

> ### Author Response · Authors · 2025-11-20
> **Response 2/2**
>
> - "*No systematic study of which types of transformations or linguistic prompts most affect model performance.*"
>
>   - **Response:** We respectfully disagree. We reported on an extensive set of models, transformations, datasets, and prompt styles to provide practical insights. We analyzed these using four metrics designed to capture different aspects of downstream usage.
>
>     Our evaluations show that no model is perfect and the ideal choice depends on the specific application. For example, Figure 4 shows that Gemini-2.0-Flash has a comparable MMScore with respect to Gemini-1.5-Pro, suggesting similar human alignment. However, if the task requires *symmetry*, Gemini-1.5-Pro is the better option because Gemini-2.0-Flash scores substantially lower. Conversely, if the task prioritizes *Smoothness*, Gemini-2.0-Flash yields a higher metric than Gemini-1.5-Pro. This indicates more variability in its generated scores, making Gemini-2.0-Flash the preferable choice for that specific requirement.
>
>     We finally note that PairBench offers tools for systematic studies when it comes to comparisons on the visual domain, and fills a gap in the literature as no other similar evaluation setup is openly available.
>
> ---
>
> - "*You mention that prompt phrasing significantly affects MMScore. Could you provide quantitative results on how large this variance is (e.g., standard deviation across prompts)?*"
>   - **Response:** Figure 7 visualizes how different prompts affect the MMScore of each model across the sub-domains of PairBench. Following the reviewer's suggestion, we have added a table to Appendix C.6 to explicitly report the variability caused by prompting. We also include this data in the table below.
> The results show that the standard deviation varies across models. Furthermore, consistent with Figure 7, no single prompt template is universally superior. This highlights the importance of using multiple prompt templates to ensure fair and robust comparisons.
> | Model | MMScore (Kendall Tau) | Exact Sym | Entropy | Controllability |
> | :--- | :---: | :---: | :---: | :---: |
> | **InternVL2.5-8B** | 81.68% ± 0.48% | 79.74% ± 0.82% | 1.404 ± 0.021 | 88.45% ± 0.47% |
> | **Qwen2.5-VL-7B** | 81.67% ± 0.51% | 77.99% ± 0.35% | 1.238 ± 0.023 | 89.13% ± 0.75% |
> | **Qwen2.5-VL-32B** | 79.22% ± 1.58% | 66.33% ± 2.12% | 1.493 ± 0.022 | 89.26% ± 0.48% |
> | **InternVL2-8B** | 77.88% ± 0.75% | 62.94% ± 6.34% | 1.293 ± 0.045 | 90.86% ± 1.86% |
> | **Qwen2-VL-7B** | 75.89% ± 0.87% | 58.24% ± 1.50% | 1.591 ± 0.034 | 90.93% ± 0.26% |
> | **GPT-4o-mini-0718** | 80.61% ± 0.75% | 56.98% ± 2.15% | 1.571 ± 0.023 | 91.57% ± 0.45% |
> | **GPT-4o-0513** | 81.03% ± 0.53% | 58.58% ± 0.94% | 1.506 ± 0.007 | 93.49% ± 0.57% |
> | **GPT-4o-0806** | 73.42% ± 3.91% | 20.25% ± 13.72% | 1.377 ± 0.047 | 83.79% ± 3.94% |
> | **GPT-4o-1120** | 83.16% ± 0.55% | 63.48% ± 0.98% | 1.494 ± 0.013 | 91.78% ± 0.53% |
> | **Gemini-1.5-Pro** | 83.84% ± 1.09% | 79.90% ± 1.40% | 1.136 ± 0.038 | 89.20% ± 1.62% |
> | **Gemini-1.5-Flash** | 82.88% ± 0.51% | 77.34% ± 1.01% | 1.314 ± 0.025 | 88.84% ± 0.62% |
> | **Gemini-2.0-Flash** | 80.74% ± 0.99% | 61.64% ± 2.59% | 1.502 ± 0.032 | 92.31% ± 0.90% |
>
> Please feel free to request further explanation if needed. If the responses sufficiently resolve the raised points, an updated score would be appreciated.

---

> ### Author Response · Authors · 2025-11-25
> **Concluding Remarks**
>
> Thank you to the reviewer for their valuable feedback. We confirm that all concerns have been fully addressed in the updated document. We look forward to hearing if any further discussion is needed.

---

> ### Comment · Reviewer_Jopj · 2025-11-26
> **Response to the rebuttal**
>
> Thank you for the detailed rebuttal and the additional analyses.
>
> * The rebutta addresses my concerns about human label reliability and provides the quantitative analysis of prompt variance.
> * However, my main concern about the limited realism of the image transformations, and the resulting mismatch between the benchmark design and the intended real-world problem settings, is not fully resolved. The authors have chosen to keep the benchmark focused on simple controlled transformations for this submission, without attempting more semantically rich or distributionally diverse comparisons.
>
> Given this, I will maintain my original overall score.

---

> > ### Author Response · Authors · 2025-11-26
> > **Response to Reviewer Jopj**
> >
> > Thank you for following up and checking our responses. We appreciate the engagement. Our interpretation from the reviewer's comment is that PairBench is valuable and fills a gap in the literature, but the reviewer would like it to offer a broader coverage. We thus note:
> >
> >
> > - Being able to control variations across images is what makes it possible to measure the metrics we report.
> >
> >
> > - Our results suggest that synthetic transformations do offer strong signals in terms of a model's ability to compare in the visual domain. We do measure specific skills (controllability, symmetry, and so on), and can even rank models in terms of much more complex downstream tasks by simply looking at a few images synthetically transformed. We'd claim that as **a feature** of PairBench as it offers an advantageous evaluation cost/informativeness ratio.
> >
> >
> > - We do report results on natural shifts, i.e., spatial transformation,via the WhatsUp images.
> >
> >
> > - **PairBench is aimed to be a live benchmark**, easily expandable, and new metrics, datasets, models, prompt templates, and inference techniques, can all be included over time. This manuscript reports on the very first iteration of the benchmark, which is already useful to the community, but it can and will grow in scope.
> >
> >
> > - On a less technical note, we'd respectfully dispute a reject score on the basis of a not large enough scope. **The benchmark, as it is, already fills a gap in the literature, provides useful information at a low cost, and thus can be very useful to the community**. It actually poses a challenge even to state of the art production grade models. The claim of a limited scope holds true for any benchmark ever published (or any submission to the conference, really), and thus should not be the basis for a disqualifying assessment. We very respectfully urge the reviewer to center their assessment around the value PairBench adds or limitations/issues it has, as no benchmark is exhaustive.

---

### Official Review · Reviewer_MYpw · 2025-10-31

**Soundness:** 2
**Presentation:** 3
**Contribution:** 2
**Rating:** 2
**Confidence:** 4

**Summary:**

This paper focuses on a critical issue of the VLM's visual comparison capability. Currently, applications of the visual comparison like automatic evaluation, retrieval re-ranking, and retrieval-augmented generation lacks systematic assessment. Existing evaluation methods either fail to isolate comparison ability from other model capabilities, rely on costly human annotation, or overlook critical flaws in state-of-the-art VLMs (e.g., score asymmetry when reversing image pair order). To address this, the author proposes PAIRBENCH, a low-cost, reproducible framework for evaluating VLMs’ visual comparison reliability. The framework’s design is anchored in two key components: (1) A controllable dataset of 70K pairs, derived from three public benchmarks (ImageNet, MS-COCO, WhatsUp), with pairs categorized as Identical (near-duplicate/images-text matches), Transformed (images with controlled manipulations like color jitter), or Irrelevant (unrelated content); ground truth (GT) scores are validated via 70+ human annotators. (2) Four complementary metrics to quantify reliability: MMScore (Kendall Tau alignment with human rankings, primary metric), ε-RelaxSym (order consistency, ε=1), Smoothness (score distribution diversity via entropy), and Controllability (responsiveness to prompt instructions). Experimental results across VLMs reveal no model dominates all metrics (e.g., GPT-5 achieves the highest MMScore (83.63%) but lower Smoothness, while InternVL2.5-8B leads open-source models (MMScore: 81.50%)). Critically, PAIRBENCH metrics (notably MMScore) exhibit strong correlations with complex reasoning benchmarks (90% with MMMU, 80% with HallusionBench).

**Strengths:**

Methodological Rigor in Dataset and Metric Design: The dataset is controllable and human-validated, which is derived from 3 public benchmarks (ImageNet, MS-COCO, WhatsUp) to ensure reproducibility, with pairs categorized into Identical/Transformed/Irrelevant to isolate specific visual variations. Ground truth scores are validated via 70+ annotators, avoiding subjective biases in "gold standard" labels.

Insights of the correlation between VLMs’ visual comparison ability and their performance on complex multimodal tasks: This insight upends the view that "visual comparison is a trivial, isolated task". Instead, it frames comparison ability as a core engine for complex tasks.

**Weaknesses:**

Disconnect between the bench’s Pair design and practical visual comparison demands: The current framework defines three Pair types (Identical/Transformed/Irrelevant) based on trivial visual manipulations (e.g., color jitter, basic rotation, Gaussian blur) or simple content overlap (e.g., near-duplicate images, random irrelevant content). However, these designs fail to capture the core real-world capabilities users actually care about: specifically, a model’s ability to (1) recognize the same object across diverse perspectives (e.g., a chair viewed from the front vs. side vs. top, with varying lighting/occlusion) and (2) establish cross-scenario consistency (e.g., identifying a cat in a studio photo vs. a street scene vs. a cartoon illustration). For example, the "Transformed" Pair in PAIRBENCH only applies superficial, single-dimensional changes (e.g., shifting hue or rotating by 90°) that rarely challenge a model’s fundamental object identity cognition. In contrast, real applications (e.g., retail product retrieval, autonomous driving object tracking, medical image diagnosis) require models to ignore viewpoint or scene variations while preserving object consistency. This "toy-like" design reduces the bench’s ecological validity: even if a model performs well on PAIRBENCH, it provides little assurance of its utility in scenarios where visual comparison truly matters. The paper also fails to justify why it prioritizes trivial manipulations over these high-stakes real-world needs, weakening the motivation for the bench’s existence.

Inappropriate statement: The paper’s core value proposition that framing PAIRBENCH as an "efficient surrogate for model selection" (via MMScore’s correlation with complex benchmarks like MMMU) suffers from flawed causal inference. The observed strong correlations (e.g., 90% between MMScore and MMMU) are not evidence of PAIRBENCH’s "representativeness," but rather likely a byproduct of a confounding variable: the model’s general visual understanding ability. A model with strong inherent visual capabilities will naturally perform well on both PAIRBENCH (which relies on basic visual similarity judgment) and complex multimodal tasks. The paper provides no evidence to rule out this alternative explanation: for instance, it does not test whether there exist models with weak general visual ability but strong PAIRBENCH performance (that then fail at complex tasks), or vice versa. Without such control experiments, the claim that PAIRBENCH is a "valid surrogate" collapses into a trivial observation: "models good at visual tasks are good at visual tasks."

**Questions:**

1. Do you have examples of models with strong general visual ability (e.g., high ImageNet accuracy) but poor PAIRBENCH performance (due to weak comparison-specific skills like order consistency)? Conversely, are there models with weak general visual ability but strong PAIRBENCH performance that still fail at complex tasks?

2. Can you provide a concrete example where a model’s PAIRBENCH performance directly predicted its success (or failure) in a non-toy visual comparison task or visual cues comparison task?

---

> ### Author Response · Authors · 2025-11-20
> **Response 1/2**
>
> Thank you for the evaluation. I respond to each of the listed weaknesses individually below.
>
> - "*Disconnect between the bench's Pair design and practical visual comparison demands:[...] why it prioritizes trivial manipulations over these high-stakes real-world needs, weakening the motivation for the bench's existence.*"
>   - **Response:** Thank you for the comment. We agree that the transformations in PairBench do not reflect the full range of real-world semantic variation. **Please note that this design is intentional**. PairBench is not a semantic similarity benchmark but a controlled diagnostic tool designed to isolate whether a model can detect specific visual changes and follow instructed sensitivities. To our knowledge, no such benchmark currently exists, and PairBench fills this gap as also noted by other reviewers (e.g., Reviewer Jopj).
>     While the transformations are controlled, we observed that performance is surprisingly lacking even for state-of-the-art models. This highlights the need for this specific type of evaluation to guide future model iterations and to help users select models for decision-making roles. PairBench therefore complements, rather than replaces, object-level or semantic-similarity evaluations.
> ---
> - "*Inappropriate statement: The paper's core value proposition that framing [...] a trivial observation: "models good at visual tasks are good at visual tasks."*"
>   - **Response:** Thank you for the comment. We only claim a correlation between MMScore and broader multimodal performance, not a causal relationship. This correlation is expected, since MMScore reflects one aspect of overall capability. However, PairBench is not reducible to general visual ability, because the other metrics capture behaviors that standard benchmarks do not measure. In practice, models, e.g., GPT4o-0853, may score highly on MMScore yet show low symmetry or less controllability compared to models with similar MMScore, revealing weaknesses that are not explained by general visual competence alone. This directly provides the counterexample the reviewer suggests is missing and shows that PairBench offers diagnostic signals beyond what existing evaluations capture.
> ---
> - "*Do you have examples of models with strong general visual ability (e.g., high ImageNet accuracy) but poor PAIRBENCH performance (due to weak comparison-specific skills like order consistency)? Conversely, are there models with weak general visual ability but strong PAIRBENCH performance that still fail at complex tasks?*"
>   - **Response:** Thank you for the question. PairBench evaluates four distinct behaviors, so the idea of a model “performing poorly” is only meaningful with respect to a specific metric. MMScore reflects general visual ability, and models with strong perception naturally score higher on this metric. However, PairBench exposes behavioral differences that are not explained by visual competence alone. For example, Figure 4 shows that GPT5 and Gemini 2.5 Pro have similarly high MMScores yet diverge across the other metrics: GPT5 is more symmetric and more controllable, while Gemini 2.5 Pro produces smoother score transitions. These contrasts demonstrate that models with comparable visual ability can behave very differently on comparison-specific tasks. Although PairBench is not designed to highlight models with weak perception, degenerate cases further illustrate that the metrics measure different behaviors: a model that always outputs “1” would have a very low MMScore but a perfect symmetry score, and a model that outputs a uniform constant across pairs would again have a low MMScore but a high smoothness score. These cases show that the four metrics capture orthogonal aspects of model behavior rather than collapsing into general visual ability.

---

> ### Author Response · Authors · 2025-11-20
> **Response 2/2**
>
> - "*Can you provide a concrete example where a model's PAIRBENCH performance directly predicted its success (or failure) in a non-toy visual comparison task or visual cues comparison task?*"
>   - **Response:** PairBench is a controlled diagnostic framework for evaluating instruction-driven comparison behavior. While our MMScore metric correlates strongly with general benchmarks, PairBench is not intended to directly predict performance on domain-specific downstream tasks. These applications typically require task-specific labels, pipelines, and definitions of similarity that are outside the scope of this work. Our goal is to provide a clean setting to assess how reliably a model compares pairs under explicit instructions, a capability directly relevant to selecting automatic judges or reward models.
>
>     To illustrate the distinction between our diagnostic metrics and real-world application, we conducted an additional experiment using Gemini-2.0-Flash and GPT-4o on images from the WhatsUp dataset. Unlike the majority of PairBench's synthetic transformations (e.g., color jitter), [these images](https://imgur.com/a/35b2OZA) contain differences where object positions have changed inside the image (i.e., Spatial transformation).We prompted the models to detect this positional change and score the similarity from 1–10. The outputs below reveal that the scores are not only asymmetric but also diverge significantly when changing the order of images, further validating the need for the controlled diagnostic setting PairBench provides. Please view the image [*here*](https://imgur.com/a/35b2OZA).
>
>
>   Prompt: ```You are tasked with evaluating the similarity between two images while paying attention to the following conditions:  - Be sensitive to spatial positions while evaluating images. For example, if the objects in the images are the same but positioned differently, the similarity score should decrease. If the objects and their positions match exactly, the similarity score should remain high. Your goal is to judge the similarity of the images overall, where satisfying the conditions increases the similarity score. If the images are identical but fail to meet any of the conditions, they should still receive a higher score than completely unrelated images. Provide a similarity score on a scale from 1 to 10, where 1 represents entirely dissimilar images and 10 represents identical images that satisfy all conditions. Ensure your response is strictly in the following format: "Score: <1-10> Reason: <reason for score>" Do not include anything else in your response. What score would you assign to this pair of images?```'''
>
> | Model | Order | Score | Reason |
> | :--- | :---: | :---: | :--- |
> | **Gemini-2.0-Flash** | img1-img2 | 7 | The images contain the same objects (a red Houston Rockets cap and a spoon) but their positions are different, causing a decrease in the similarity score. |
> | **Gemini-2.0-Flash** | img2-img1 | 3 | The images contain the same objects (hat and spoon) but the spatial position of these objects differs substantially, as does the orientation of the hat. |
> | **GPT-4o** | img1-img2 |7 | The two images contain the same objects (a red cap and a metal spoon), but their spatial positions are reversed: in the first image the cap is in front and the spoon is farther back, while in the second the spoon is in front and the cap is farther back. The content is identical, but the positional change lowers the similarity. |
> | **GPT-4o** | img2-img1 | 4 | Both images contain the same two objects—a red cap and a metal spoon—but their spatial positions are reversed. In the first image, the spoon is in the foreground and the cap is in the background, while in the second image the cap is in the foreground and the spoon is in the background. This significant positional change reduces similarity, though the shared objects keep the score above very low values. |
>
> We will be available to clarify any remaining issues. If the provided responses address your concerns, we would appreciate a revised rating.

---

> ### Author Response · Authors · 2025-11-25
> **Concluding Remarks**
>
> We appreciate the time the reviewer took to review our submission. We have carefully responded to all points raised in their assessment. Should the reviewer have any follow-up questions or require further clarification on the revisions, please do not hesitate to reach out.

---

### Official Review · Reviewer_a2Ys · 2025-11-01

**Soundness:** 2
**Presentation:** 2
**Contribution:** 1
**Rating:** 2
**Confidence:** 4

**Summary:**

The paper introduces PAIR Bench, where there used this to systematically assess vlms ability to make reliable judgements over data pairs(image-text, image-image). They used off-the shelf datasets, a set of controlled image transformations and human-annotated ground truth to create the benchmark and then used four key evaluation metrics.

**Strengths:**

- The paper tackles one issue: As people are using more and more VLMS as a judge, how trustworthy it is as an automated evaluator or rankers.
- The benchmark uses existing datasets with controlled transformers. They conducted a thorough evaluation of multiple VLMs. Using different prompt to evaluate was a good point as well.
- The paper is easy to read.

**Weaknesses:**

- Having a benchmark on pairwise images and VLMs that fail to recognise it, is hardly new. [1][2]
- The method merely uses basic data augmentations to create the benchmark. These do not represent what users use VLM in real life to compare or judge between samples. These are mostly synthetic or self-generated pairings. The task seems to be more of a visual robustness test.
- A proper ablation study of why models fail, whether it is a vision encoder problem, or the alignment problem, is needed. Chain of thought prompting or in-context learning should also help models.
- Feature analysis on the vision foundation models or cosine similarity of features from [maybe from florence, coca] those two images, would be good ablation study. It would add some value as it's a perception problem or language prompting problem.
- The paper claims that high correlations between pairbench metrics and multimodal benchmark performance. That means PAIRbench is not unique.
- More information on how dataset spilt was used from an shelf dataset is needed.


[1] ReMI: A Dataset for Reasoning with Multiple Images
[2] MLLM-CompBench: A Comparative Reasoning Benchmark for Multimodal LLMs

**Questions:**

- I fail to understand the novelty of the benchmark. Author may further clarify why this is unique relating with real-world scenario,

---

> ### Author Response · Authors · 2025-11-20
> **Response 1/2**
>
> Thank you for the evaluation. I respond to each of the listed weaknesses individually below.
>
> - "*Having a benchmark on pairwise images and VLMs that fail to recognise it, is hardly new. [1][2]. The method merely uses basic data augmentations to create the benchmark. These do not represent what users use VLM in real life to compare or judge between samples. These are mostly synthetic or self-generated pairings. The task seems to be more of a visual robustness test.*"
>   - **Response:** Thank you for pointing out these works. We will cite them. We do not claim to introduce the first pairwise or multi-image evaluation. As also acknowledged by reviewer Jopj, the novelty of PairBench lies in its controlled and instruction-driven design, which uses simple and well-defined transformations to avoid complex reasoning and to create an interpretable setting for evaluating whether models can detect specific changes, follow explicit similarity criteria, and produce consistent scores. Although the pairs are synthetic, this capability is essential for selecting reliable VLM judges or reward models, where precise and instruction-aligned comparisons are required. In addition, existing works do not provide structured metrics such as symmetry, smoothness, and controllability, which reveal behavioral differences that standard accuracy-style measures overlook and which give practitioners a more nuanced tool for choosing models based on the specific properties their applications require.
> ---
> - "*A proper ablation study of why models fail, whether it is a vision encoder problem, or the alignment problem, is needed. Chain of thought prompting or in-context learning should also help models.*"
>   - **Response:** We appreciate the suggestion. A detailed ablation of failure sources, such as separating vision encoder limitations from alignment issues, would indeed be valuable. However, our primary goal is to evaluate which VLMs function most reliably as automatic judges, and this setting is dominated by closed-source models where component-level analysis is not possible. As a result, we intentionally treat all models as black-box judges, and a full diagnostic breakdown lies outside the scope of this work. Regarding chain-of-thought or in-context prompting, we explored multiple prompt templates and observed that using several templates improves stability and robustness. More advanced prompting strategies, as well as deeper ablations for models where this is feasible, are promising directions for future work.
> ---
> - "*Feature analysis on the vision foundation models or cosine similarity of features from [maybe from florence, coca] those two images, would be good ablation study. It would add some value as it's a perception problem or language prompting problem.*"
>   - **Response:** We thank the reviewer for their comment. Note that the aim of PairBench is to provide the community with evaluation tools (data and metrics) and a systematic framework for picking automatic evaluators, whether as judges or reward models. We do agree that a natural next step, given meaningful evaluation protocols such as what we provide, is to identify the reasons why models fail, and we also agree that inspecting visual representations can lead to explanations. This is an interesting future direction given the variety of models observed over the different metrics in PairBench and how the best performing models on MMScore do not necessarily have the highest symmetries or vary on how they compare with each other on smoothness and controllability.
> ---
> - "*The paper claims that high correlations between pairbench metrics and multimodal benchmark performance. That means PAIRbench is not unique.*"
>   - **Response:** Thank you for the comment. We would like to clarify that only one of our four metrics, the MMScore, shows a strong correlation with general multimodal benchmark performance, which simply indicates that MMScore captures a component of overall model quality. The remaining three metrics, symmetry, smoothness, and controllability, do not correlate strongly with standard benchmarks because they measure behavioral properties that existing evaluations do not track. These metrics expose how consistently a model judges pairs, how stable its scoring behavior is, and whether it follows instructed invariances. PairBench provides a unified framework for measuring all four aspects, and these properties are crucial when selecting reliable automatic judges or reward models, which require more than raw benchmark accuracy alone.

---

> ### Author Response · Authors · 2025-11-20
> **Response 2/2**
>
> - "*More information on how dataset spilt was used from an shelf dataset is needed.*"
>   - **Response:** We provide full details of how the off-the-shelf datasets were split and used in Appendix D, and in particular in Section D.1 on dataset creation. The procedure, selection criteria, and construction of the split are all described there. If any specific aspect of the split remains unclear, we would be happy to clarify it and update the manuscript accordingly.
>     The full data with the exact splits we used and all the transformations we applied will be released upon publication.
> ---
> - "*I fail to understand the novelty of the benchmark. Author may further clarify why this is unique relating with real-world scenario,*"
>   - **Response:** Thank you for the comment. PairBench introduces a simple and controlled framework for evaluating how reliably a VLM can compare image-image and image-text pairs under explicit instructions. To our knowledge, no such benchmark currently exists, and PairBench fills this gap, as also noted by other reviewers (e.g., Reviewer Jopj). It defines four structured metrics that capture different aspects of model behavior. The evaluation protocol we introduced helped uncover important limitations in state-of-the-art models, and the evaluations PairBench introduced will inform the next iteration of models so we can get models that more reliably parse visual information.
>     Our choice for using synthetic pairs is so that controlled transformations allow us to isolate core requirements for real-world uses like automatic judges and reward models. Standard multimodal benchmarks do not test these properties or provide metrics like symmetry, smoothness, and controllability. Crucially, we find that performance is surprisingly lacking even for state-of-the-art models. This highlights the need for this kind of evaluation to inform the next iterations of models.
>
> If additional detail is needed, I welcome further questions. If the responses resolve the concerns, I would highly appreciate it if you consider updating your score.

---

> ### Author Response · Authors · 2025-11-25
> **Concluding Remarks**
>
> We appreciate the reviewer's insightful comments, which helped improve the quality of this work. All raised concerns have been thoroughly addressed in the revised manuscript. We welcome any further feedback or questions the reviewer may have upon reviewing the changes.

---

### Official Review · Reviewer_JkZh · 2025-11-01

**Soundness:** 2
**Presentation:** 4
**Contribution:** 3
**Rating:** 4
**Confidence:** 3

**Summary:**

The authors propose PairBench, a benchmark for measuring the capability of Vision-Language Models as similarity kernels. PairBench is composed of image-image and image-text pairs that belong to one of three groups: (i) identical pairs, featuring almost the same image (or corresponding caption); (ii) transformed pairs, involving the transformation of an image with one of six transformations and (iii) irrelevant pairs, where the images are distinct. The task is to assign a similarity score to the pair under two conditions: one where the model should be invariant to a specified transformation and another where it should be sensitive. Model similarity scores are then assessed with four separate metrics measuring correlation with ground-truth judgements, symmetry, controllability and smoothness of the similarity distribution. The authors build on top of pre-existing datasets to construct PairBench and evaluate a broad range of vision-language models on their benchmark, finding variability in rankings of models for different metrics.

**Strengths:**

- The four metrics considered are each well-motivated for the task of evaluating models as similarity kernels.
- The authors' evaluation of a broad range of models with different sizes, different strengths and different access (open/proprietary) allows them to describe interesting trends in task performance. Each of the evaluations are made more robust through the inclusion of multiple prompts.
- The insights regarding the lack of similarity are important, as they directly affect the cost-effectiveness of LLM judges for the task of similarity estimation. I found the relatively poor performance of the Gemini models surprising.
- The paper is clearly written throughout.

**Weaknesses:**

- My main criticism, and the reason for the soundness score and the overall rating, is the nature of the transformations. The only images that can be given a high ground-truth similarity score (10 or 6, from the fixed set of scores) are identical or transformed images. In real applications, however, practitioners are likely to be computing the similarity of distinct images (including those found in Figure 16 that are not identical but could be considered similar due to them both containing birds). Treating all distinct images as maximally dissimilar regardless of semantic content makes it difficult for me to see how insights from the benchmark can be generalized. Though perhaps I am missing something in the pair construction or the ground-truth score assignment.
- As a corollary, setting ground-truth similarity scores to only 3 distinct values could affect the validity of MMScores. I would be curious to see how model correlations with judgements change when the 300 image pairs in the human study are used alongside the actual human judgements themselves.
- I am not sure whether the claim in the introduction regarding reasoning performance depending on models functioning as effective similarity kernels holds. These two values are correlated, but that should not imply causation. From what I can tell, model performance on PairBench is correlated with the model's overall quality, as more recent and larger models tend to achieve higher MMScores.
- It is unclear to me how to interpret the smoothness or the controllability results, as model performances are tightly clustered around similar values.

**Questions:**

- Just to make sure that my understanding is correct: For identical and transformed cases, the second is image is always based on the first image, correct? Furthermore, my understanding is that the ground-truth similarity score is based purely on the pair category and the prompt type. Would that be the right interpretation? That is at least my reading of the paper, but Figure 16 makes me doubt my understanding.

---

> ### Author Response · Authors · 2025-11-20
> **Response 1/2**
>
> Thank you for your thoughtful and detailed review. I appreciate your constructive feedback, and below I respond to each of your concerns point by point.
> - "*My main criticism, and the reason for the soundness score and the [...] generalized. Though perhaps I am missing something in the pair construction or the ground-truth score assignment.*"
>   - **Response**: Thank you for the comment. Note that PairBench is intentionally designed as a controlled diagnostic tool rather than a semantic similarity dataset. The aim is to cleanly evaluate how well models detect specific transformations, follow instructions about invariance or sensitivity, and translate that instructed behavior into consistent similarity scores. Those capabilities are impossible to assess using semantically similar but distinct images. PairBench complements, not replaces, broader semantic similarity benchmarks.
>
>     Also note that we do not assume all distinct images are inherently maximally dissimilar; in this benchmark, they serve as stable anchors for testing instruction-driven behavior under controlled conditions. We will clarify this motivation in the paper and plan to explore incorporating semantic similarity tasks in future work.
> ---
> - "*As a corollary, setting ground-truth similarity scores to only 3 distinct values could affect [...] used alongside the actual human judgements themselves.*"
>   - **Response:** Thank you for the comment. We believe that correlating model scores directly with human judgments in this context would beat the purpose of PairBench. Our specific goal is to avoid injecting semantic ambiguity or requiring costly manual annotation. We instead focus on controlled, interpretable test cases where the ground truth is unambiguous and defined by the transformation itself.
>     While the ground truth consists of discrete values, this design provides a rigorous and meaningful test of instruction following. It evaluates if a model can distinguish between distinct, objective states (such as identical versus transformed). Since we find that models struggle to distinguish even these clear categories, this metric successfully captures a fundamental limitation in current capabilities that subjective human correlation might obscure."
> ---
>
> - "*I am not sure whether the claim in the introduction regarding reasoning performance depending [...] more recent and larger models tend to achieve higher MMScores.*"
>   - **Response:** Thank you for raising this point. We agree that PairBench does not establish a causal link between MMScores and overall reasoning performance, and we adjusted the phrasing to ensure our claim is clear. Comparing, contrasting, and relating inputs under instructions is a key component of many reasoning tasks. PairBench isolates this specific capability in a controlled manner.
> While MMScores do correlate with model size and recency, **this pattern does not hold uniformly across all PairBench metrics**. For example, in the smoothness metric, smaller models can behave as smoothly or as biased as much larger models. Likewise, on symmetry, proprietary models (e.g., Gemini 1.5 Pro vs Gemini 2.0 Flash) with similar overall MMScore capability diverge sharply. These discrepancies highlight that PairBench is not simply measuring model scale, but is revealing distinct behavioral differences in how models implement instruction-driven similarity judgments. We will clarify this nuance and the non-causal scope of our claim in the introduction.
> ---
> - "*It is unclear to me how to interpret the smoothness or the controllability results, as model performances are tightly clustered around similar values.*"
>   - **Response:** The idea is to show that models act differently regarding smoothness and controllability; these metrics are not meaningful by themselves, but they are used to *compare* whether a model is more/less smooth or controllable than another model on a specific task or regarding a specific transformation feature.
> ---

---

> ### Author Response · Authors · 2025-11-20
> **Response 2/2**
>
> - "*Just to make sure that my understanding is correct: For identical and transformed cases, the second is image is always based on the first image, correct? [...] but Figure 16 makes me doubt my understanding.*"
>
>   - **Response:** Your understanding is correct. The ground-truth similarity score is determined solely by the pair category and the prompt type. Identical pairs always receive a score of 10. Transformed pairs, where the second image is produced by applying a specific transformation to the first, receive either 10 or 6 depending on whether the prompt instructs the model to be invariant or sensitive. Irrelevant pairs receive a score of 1. This label indicates only that the images differ and does not imply anything about semantic content. This design ensures that the benchmark focuses strictly on transformation recognition and instruction following without introducing semantic ambiguity.
>
>
>     Regarding Figure 16, even if both images happen to depict birds, the pair is still labeled irrelevant and therefore has a ground-truth score of 1. We added a short clarifying note near this figure to avoid confusion about semantically similar but irrelevant examples. We also plan to explore the inclusion of semantic similarity tasks in future work to complement this controlled setting.
>
> If any clarification is still required, I invite further engagement. If the responses adequately address your concerns, I would appreciate an updated assessment.

---

> ### Author Response · Authors · 2025-11-25
> **Concluding Remarks**
>
> We thank the reviewer for their careful assessment of our work. We have addressed all concerns raised, and we believe the revisions significantly strengthen the manuscript. We remain available and happy to continue the discussion should any new questions arise. Thank you.

---

### Author Response · Authors · 2025-11-20

Across all reviews, we appreciate the consistent recognition of the core strengths of our work. Reviewers acknowledged our motivation to evaluate VLMs as reliable judges, as Reviewer a2Ys noted that “the paper tackles one issue: as people are using more and more VLMs as a judge, how trustworthy it is as an automated evaluator.” They also appreciated the clarity and accessibility of the writing, reflected in Reviewer JkZh’s comment that “*the paper is clearly written throughout*” and Reviewer a2Ys’s statement that “*the paper is easy to read*.” The methodological rigor of our controllable, human-validated dataset and evaluation design was highlighted by Reviewer MYpw, who emphasized that the dataset is “*controllable and human-validated… to ensure reproducibility*,” and by Reviewer JkZh, who noted that “*the four metrics considered are each well-motivated*.” Reviewers also praised the breadth of our empirical study and the insights it revealed, as JkZh remarked that our evaluation “*allows them to describe interesting trends*” and that our findings on Gemini “*were surprising*,” while Reviewer Jopj noted that PairBench “*fills a clear gap in multimodal evaluation by focusing on pairwise relational reliability*” and that our correlation analysis “*strengthens the empirical insight that comparison ability predicts higher-level reasoning*.” **Collectively, these comments affirm that the motivation, design, and empirical contributions of PairBench were well-received.**

Below we summarize the shared concerns raised across reviews and explain how we have addressed them.

- Concern 1: Transformations are too simple and lack semantic depth

  - **Response:** PairBench is intentionally designed as a simple and low-cost benchmark, able to provide useful signal in terms of model behavior in comparison scenarios. This focus enables us to isolate specific comparison behaviors like symmetry, smoothness, and controllability to generate unambiguous ground truth. While we agree that including a wider range of transformations is beneficial, our results indicate that existing models already struggle significantly with these synthetic variations. As such, these transformations are sufficient to provide a clear signal of the properties we aim to measure.


- Concern 2: Reliability and subjectivity of human judgments

  - **Response:** We computed Krippendorff’s alpha (0.9396) and leave-one-rater-out Spearman correlations (0.9166) using 70+ annotators, confirming that human similarity judgments for PairBench are stable and reliable. We added these results to Appendix D.2.

- Concern 3: Correlations with external benchmarks may imply overclaimed representativeness

  - **Response:** We clarified that our correlations are descriptive and not causal, and reworded the manuscript to clarify our claim. Only MMScore aligns with general capability, while the other metrics reveal behaviors not explained by visual skill alone. For example, models with similar MMScores (e.g., GPT5 and Gemini 2.5 Pro) diverge substantially in symmetry, controllability, or smoothness. These distinctions demonstrate that *PairBench evaluates orthogonal behavioral properties essential for model selection in judging or reward-model settings*.

- Concern 4: Scope and validity of PairBench

  - **Response:** We highlight that our focus is to design a cost-efficient, controllable, and simple evaluation suite that provides clear insight in model behavior in comparison scenarios in the visual domain. In other words, the purpose of PairBench is to evaluate whether a model can perform reliable, instruction-driven comparisons under controlled conditions. Pairbench is not intended to model all real-world comparison scenarios, And our design aligns with relevant applications such as automatic judging and reward modeling, where consistency and instruction-following behavior are crucial. Studying how models selected via PairBench behave in downstream settings is a valuable direction for future work, but falls outside the scope of this benchmark. We highlight that PairBench enabled to uncover undesired behaviors even in state-of-the-art models, such as lack of basic symmetry properties.


We thank the reviewers again for their thoughtful assessments and constructive feedback. We believe PairBench addresses an important and growing need in evaluating VLMs as reliable judges, and we are committed to refining and extending the benchmark as models advance. We hope this work will serve the research community by offering a clear, reproducible, and behaviorally grounded framework for understanding and improving comparative reasoning in multimodal models, and we welcome continued discussion and engagement from the community.

---

### Comment · Area_Chair_4D3k · 2025-11-25
**Discussion Period**

Dear Reviewers and Authors,

Thank you to the authors for submitting your rebuttal. We kindly encourage reviewers to take a moment to read the response and share any follow-up thoughts. Your timely engagement at this stage is highly valuable and helps ensure a fair, well-informed final decision.

We appreciate everyone’s efforts and contributions to the process.

Warm regards, Your AC

---

### Meta-Review · Area_Chair_yCVE · 2026-01-06

**Summary:**

This paper introduces PairBench, a benchmark designed to evaluate the reliability of vision–language models (VLMs) in pairwise comparison tasks. The benchmark constructs controlled image–image and image–text pairs using simple transformations and proposes four metrics (alignment with human judgment, symmetry, smoothness, and controllability) to analyze model behavior. Extensive experiments are conducted across a wide range of open- and closed-source VLMs, revealing systematic inconsistencies in comparison behavior and correlations with existing multimodal benchmarks.

Reviewers raised several decision-relevant concerns. These include the `limited novelty and scope of the benchmark`, with questions about `whether pairwise comparison under simple, synthetic transformations meaningfully advances beyond prior comparison or robustness benchmarks`. Reviewers also expressed concerns about `the design of ground-truth similarity scores and metrics`, noting that `the use of discrete labels and fixed pair categories may limit interpretability and real-world relevance`. In addition, reviewers questioned `the generalizability and practical implications of the benchmark`, including whether it primarily measures instruction-following or visual robustness rather than comparison ability, and whether the observed correlations reflect unique insights beyond overall model quality.

**Reviewer Concerns:**

**1. [Outstanding] Limited realism and scope of pair construction (by: `JkZh`, `a2Ys`, `MYpw`, `Jopj`)**

All the reviewers argued that the benchmark relies primarily on simple, synthetic image transformations (e.g., rotation, color jitter, blur) and treats all non-identical image pairs as maximally dissimilar. They noted that such design focuses on low-level invariances rather than semantic or object-level similarity, and therefore may not reflect realistic visual comparison scenarios encountered in practice (e.g., cross-view, cross-domain, or stylistically diverse images).

The rebuttal clarified that PairBench is intentionally designed as a controlled diagnostic benchmark rather than a semantic similarity benchmark, and emphasized interpretability and low annotation cost. However, reviewers (notably `Jopj`) maintained that this design choice leaves a mismatch between the benchmark setting and real-world comparison tasks, and this concern was not fully resolved.

**2. [Partially addressed] Ground-truth similarity design and metric validity (by: `JkZh`, `a2Ys`, `Jopj`)**

Reviewers questioned the ground-truth similarity assignment, which is restricted to a small set of discrete values determined solely by pair category and prompt type. They expressed concerns that this discretization may limit the interpretability and statistical validity of correlation-based metrics such as MMScore, and that assuming such labels as a gold standard may oversimplify similarity judgments.

The rebuttal provided additional justification for using discrete, unambiguous ground truth and reported high inter-annotator agreement to support label reliability. While these clarifications address annotation consistency, concerns remain about whether the labeling scheme meaningfully captures nuanced similarity behavior relevant to broader use cases.

**3. [Partially addressed] Over-interpretation of correlations with external benchmarks (by: `JkZh`, `a2Ys`, `MYpw`)**

Several reviewers cautioned that the reported strong correlations between PairBench metrics (especially MMScore) and other multimodal benchmarks may be over-interpreted. They argued that such correlations could largely reflect overall model scale or general visual capability, rather than a distinct comparison-specific skill, and that the paper does not sufficiently rule out these alternative explanations.

The rebuttal clarified that the correlations are descriptive rather than causal and emphasized that other metrics (e.g., symmetry, smoothness, controllability) capture orthogonal behaviors. Nonetheless, it is still unconvincing that PairBench uniquely measures comparison ability beyond general model quality.

**4. [Outstanding] Limited diagnostic insight into failure sources (by: `a2Ys`)**

One reviewer noted that the benchmark treats models as black boxes and does not attempt to disentangle whether observed failures arise from the vision encoder, language alignment, or prompting strategy. The reviewer suggested that additional analyses (e.g., feature-level comparisons or controlled prompting studies) would be necessary to better understand why models fail.

The rebuttal acknowledged this limitation and framed such analyses as out of scope, leaving this concern unresolved.

**5. [Addressed] Reliability of human annotations (by: `Jopj`)**

One reviewer questioned whether human similarity judgments are sufficiently reliable to serve as ground truth.

The rebuttal directly addressed this concern by reporting Krippendorff's alpha and leave-one-rater-out analyses, demonstrating high inter-annotator consistency. This concern was resolved.

**6. [Partially addressed] Unclear practical implications and downstream relevance (by: `MYpw`, `Jopj`)**

Reviewers questioned whether performance on PairBench meaningfully predicts a model's usefulness in practical downstream comparison tasks, noting the lack of concrete demonstrations connecting benchmark results to non-synthetic, real-world applications. They expressed uncertainty about how the benchmark should be used for model selection beyond the controlled setting studied.

While the rebuttal argued that PairBench is intended as a low-cost diagnostic tool for selecting reliable judges or reward models, the practical impact and external validity remain unclear.

**Reviewer Scores:**

**Reviewer JkZh (4 -> 4)**

This reviewer considered the paper marginally below the acceptance threshold, with core concerns about the realism of the pair construction, discrete ground-truth design, and interpretation of correlations with other benchmarks. While the rebuttal clarified the authors' intent and improved framing, it did not fundamentally change the reviewer's view on generalizability and practical relevance. The score would likely remain unchanged.

**Reviewer a2Ys (2 -> 2)**

This reviewer raised a strong rejection regarding novelty, realism, and real-world relevance, viewing the benchmark as closer to a synthetic robustness test than a meaningful comparison benchmark. Although the rebuttal addressed annotation reliability and scope clarification, the central concerns remain unresolved. From the AC perspective, no score change is expected.

**Reviewer Jopj (3 -> 4)**

This reviewer acknowledged that PairBench fills a gap in evaluating pairwise comparison reliability, but expressed reservations about scope and downstream impact. The rebuttal convincingly addressed concerns about annotation reliability and clarified the intended diagnostic role of the benchmark. While limitations regarding external validity remain, these clarifications could reasonably lead to a modest increase in score, though still below the acceptance threshold.

**Reviewer MYpw (5 -> 5)**

This reviewer appreciated the controlled design and empirical analysis but questioned the benchmark's practical implications and diagnostic depth. The rebuttal improved clarity but did not provide additional evidence linking benchmark performance to real-world comparison tasks. The overall assessment would likely remain stable.

---

### Decision · Program_Chairs · 2026-01-26

Reject